# Isometric Transformation Invariant and Equivariant Graph Convolutional Networks

**Masanobu Horie**
University of Tsukuba,
Research Institute for Computational Science Co. Ltd.
`horie@ricos.co.jp`

**Naoki Morita**
University of Tsukuba,
Research Institute for Computational Science Co. Ltd.
`morita@ricos.co.jp`

**Toshiaki Hishinuma & Yu Ihara**
Research Institute for Computational Science Co. Ltd.
`{hishinuma,ihara}@ricos.co.jp`

**Naoto Mitsume**
University of Tsukuba
`mitsume@kz.tsukuba.ac.jp`

## Abstract

Graphs are one of the most important data structures for representing pairwise relations between objects. Specifically, a graph embedded in a Euclidean space is essential to solving real problems, such as physical simulations. A crucial requirement for applying graphs in Euclidean spaces to physical simulations is learning and inferring the isometric transformation invariant and equivariant features in a computationally efficient manner. In this paper, we propose a set of transformation invariant and equivariant models based on graph convolutional networks, called IsoGCNs. We demonstrate that the proposed model has a competitive performance compared to state-of-the-art methods on tasks related to geometrical and physical simulation data. Moreover, the proposed model can scale up to graphs with 1M vertices and conduct an inference faster than a conventional finite element analysis, which the existing equivariant models cannot achieve.

## 1 Introduction

Graph-structured data embedded in Euclidean spaces can be utilized in many different fields such as object detection, structural chemistry analysis, and physical simulations. Graph neural networks (GNNs) have been introduced to deal with such data. The crucial properties of GNNs include permutation invariance and equivariance. Besides permutations, isometric transformation invariance and equivariance must be addressed when considering graphs in Euclidean spaces because many properties of objects in the Euclidean space do not change under translation and rotation. Due to such invariance and equivariance, 1) the interpretation of the model is facilitated; 2) the output of the model is stabilized and predictable; and 3) the training is rendered efficient by eliminating the necessity of data augmentation as discussed in the literature (Thomas et al., 2018; Weiler et al., 2018; Fuchs et al., 2020).

Isometric transformation invariance and equivariance are inevitable, especially when applied to physical simulations, because every physical quantity and physical law is either invariant or equivariant to such a transformation. Another essential requirement for such applications is computational efficiency because the primary objective of learning a physical simulation is to replace a computationally expensive simulation method with a faster machine learning model.

In the present paper, we propose *IsoGCNs*, a set of simple yet powerful models that provide computationally-efficient isometric transformation invariance and equivariance based on graph convolutional networks (GCNs) (Kipf & Welling, 2017). Specifically, by simply tweaking the definition of an adjacency matrix, the proposed model can realize isometric transformation invariance. Because the proposed approach relies on graphs, it can deal with the complex shapes that are usually presented using mesh or point cloud data structures. Besides, a specific form of the IsoGCN layer can be regarded as a spatial differential operator that is essential for describing physical laws. In addition, we have shown that the proposed approach is computationally efficient in terms of processing graphs

with up to 1M vertices that are often presented in real physical simulations. Moreover, the proposed model exhibited faster inference compared to a conventional finite element analysis approach at the same level of accuracy. Therefore, an IsoGCN can suitably replace physical simulations regarding its power to express physical laws and faster, scalable computation. The corresponding implementation and the dataset are available online[1].

The main contributions of the present paper can be summarized as follows:

- We construct isometric invariant and equivariant GCNs, called IsoGCNs for the specified input and output tensor ranks.
- We demonstrate that an IsoGCN model enjoys competitive performance against state-of-the-art baseline models on the considered tasks related to physical simulations.
- We confirm that IsoGCNs are scalable to graphs with 1M vertices and achieve inference considerably faster than conventional finite element analysis.

## 2 RELATED WORK

**Graph neural networks.** The concept of a GNN was first proposed by Baskin et al. (1997); Sperduti & Starita (1997) and then improved by (Gori et al., 2005; Scarselli et al., 2008). Although many variants of GNNs have been proposed, these models have been unified under the concept of message passing neural networks (Gilmer et al., 2017). Generally, message passing is computed with nonlinear neural networks, which can incur a tremendous computational cost. In contrast, the GCN developed by Kipf & Welling (2017) is a considerable simplification of a GNN, that uses a linear message passing scheme expressed as

$$\boldsymbol{H}_{\text{out}} = \sigma(\hat{\boldsymbol{A}}\boldsymbol{H}_{\text{in}}\boldsymbol{W}), \tag{1}$$

where $\boldsymbol{H}_{\text{in}}$ ($\boldsymbol{H}_{\text{out}}$) is an input (output) feature of the $l$th layer, $\hat{\boldsymbol{A}}$ is a renormalized adjacency matrix with self-loops, and $\boldsymbol{W}$ is a trainable weight. A GCN, among the variants of GNNs, is essential to the present study because the proposed model is based on GCNs for computational efficiency.

**Invariant and equivariant neural networks.** A function $f : X \rightarrow Y$ is said to be equivariant to a group $G$ when $f(g \cdot x) = g \cdot f(x)$, for all $g \in G$ and $x \in X$, assuming that group $G$ acts on both $X$ and $Y$. In particular, when $f(g \cdot x) = f(x)$, $f$ is said to be invariant to the group $G$. Group equivariant convolutional neural networks were first proposed by Cohen & Welling (2016) for discrete groups. Subsequent studies have categorized such networks into continuous groups (Cohen et al., 2018), three-dimensional data (Weiler et al., 2018), and general manifolds (Cohen et al., 2019). These methods are based on CNNs; thus, they cannot handle mesh or point cloud data structures as is. Specifically, 3D steerable CNNs (Weiler et al., 2018) uses voxels (regular grids), which though relatively easy to handle, are not efficient because they represent both occupied and non-occupied parts of an object (Ahmed et al., 2018). In addition, a voxelized object tends to lose the smoothness of its shape, which can lead to drastically different behavior in a physical simulation, as typically observed in structural analysis and computational fluid dynamics.

Thomas et al. (2018); Kondor (2018) discussed how to provide rotation equivariance to point clouds. Specifically, the tensor field network (TFN) (Thomas et al., 2018) is a point cloud based rotation and translation equivariant neural network the layer of which can be written as

$$\tilde{\mathbf{H}}_{\text{out},i}^{(l)} = w^{ll} \, \tilde{\mathbf{H}}_{\text{in},i}^{(l)} + \sum_{k \geq 0} \sum_{j \neq i} \boldsymbol{W}^{lk}(\boldsymbol{x}_j - \boldsymbol{x}_i) \, \tilde{\mathbf{H}}_{\text{in},j}^{(k)}, \tag{2}$$

$$\boldsymbol{W}^{lk}(\boldsymbol{x}) = \sum_{J=|k-l|}^{k+l} \phi_J^{lk}(\|\boldsymbol{x}\|) \sum_{m=-J}^{J} Y_{Jm}(\boldsymbol{x}/\|\boldsymbol{x}\|)\boldsymbol{Q}_{Jm}^{lk}, \tag{3}$$

where $\tilde{\mathbf{H}}_{\text{in},i}^{(l)}$ ($\tilde{\mathbf{H}}_{\text{out},i}^{(l)}$) is a type-$l$ input (output) feature at the $i$th vertex, $\phi_J^{lk} : \mathbb{R}_{\geq 0} \rightarrow \mathbb{R}$ is a trainable function, $Y_{Jm}$ is the $m$th component of the $J$th spherical harmonics, and $\boldsymbol{Q}_{Jm}^{lk}$ is the Clebsch-Cordan coefficient. The SE(3)-Transformer (Fuchs et al., 2020) is a variant of the TFN with self-attention. These models achieve high expressibility based on spherical harmonics and message passing with nonlinear neural networks. However, for this reason, considerable computational resources

---

[1]https://github.com/yellowshippo/isogcn-iclr2021

are required. In contrast, the present study allows a significant reduction in the computational costs because it eliminates spherical harmonics and nonlinear message passing. From this perspective, IsoGCNs are also regarded as a simplification of the TFN, as seen in equation 14.

**Physical simulations using GNNs.** Several related studies, including those by Sanchez-Gonzalez et al. (2018; 2019); Alet et al. (2019); Chang & Cheng (2020) focused on applying GNNs to learn physical simulations. These approaches allowed the physical information to be introduced to GNNs; however, addressing isometric transformation equivariance was out of the scope of their research.

In the present study, we incorporate isometric transformation invariance and equivariance into GCNs, thereby, ensuring the stability of the training and inference under isometric transformation. Moreover, the proposed approach is efficient in processing large graphs with up to 1M vertices that have a sufficient number of degrees of freedom to express complex shapes.

## 3 ISOMETRIC TRANSFORMATION INVARIANT AND EQUIVARIANT GRAPH CONVOLUTIONAL LAYERS

In this section, we discuss how to construct IsoGCN layers that correspond to the isometric invariant and equivariant GCN layers. To formulate a model, we assume that: 1) only attributes associated with vertices and not edges; and 2) graphs do not contain self-loops. Here, $\mathcal{G} = (\mathcal{V}, \mathcal{E})$ denotes a graph and $d$ denotes the dimension of a Euclidean space. In this paper, we refer to tensor as geometric tensors, and we consider a (discrete) rank-$p$ tensor field $\mathbf{H}^{(p)} \in \mathbb{R}^{|\mathcal{V}| \times f \times d^p}$, where $|\mathcal{V}|$ denotes the number of vertices and $f \in \mathbb{Z}^+$ ($\mathbb{Z}^+$ denotes the positive integers). Here, $f$ denotes the number of features (channels) of $\mathbf{H}^{(p)}$, as shown in Figure 1 (a). With the indices, we denote $H^{(p)}_{i;g;k_1 k_2 \ldots k_p}$, where $i$ permutes under the permutation of vertices and $k_1, \ldots, k_p$ refers to the Euclidean representation. Thus, under the permutation, $\pi : H^{(p)}_{i;g;k_1 k_2 \ldots k_p} \mapsto H^{(p)}_{\pi(i);g;k_1 k_2 \ldots k_p}$, and under orthogonal transformation, $\boldsymbol{U} : H^{(p)}_{i;g;k_1 k_2 \ldots k_p} \mapsto \sum_{l_1, l_2, \ldots, l_p} U_{k_1 l_1} U_{k_2 l_2} \ldots U_{k_p l_p} H^{(p)}_{i;g;l_1 l_2 \ldots l_p}$.

### 3.1 CONSTRUCTION OF AN ISOMETRIC ADJACENCY MATRIX

Before constructing an IsoGCN, an **isometric adjacency matrix** (IsoAM), which is at the core of the IsoGCN concept must be defined. The proof of each proposition can be found in Appendix B.

An IsoAM $\mathbf{G} \in \mathbb{R}^{|\mathcal{V}| \times |\mathcal{V}| \times d}$ is defined as:

$$\mathbb{R}^d \ni \mathbf{G}_{ij;;:} := \sum_{k,l \in \mathcal{V}, k \neq l} \boldsymbol{T}_{ijkl}(\boldsymbol{x}_k - \boldsymbol{x}_l), \quad (4)$$

where $\mathbf{G}_{ij;;:}$ is a slice in the spatial index of $\mathbf{G}$, $\boldsymbol{x}_i \in \mathbb{R}^d$ is the position of the $i$th vertex (rank-1 tensor), and $\boldsymbol{T}_{ijkl} \in \mathbb{R}^{d \times d}$ is an untrainable transformation invariant and orthogonal transformation equivariant rank-2 tensor. Note that we denote $\mathbf{G}_{ij;;k}$ to be consistent with the no-

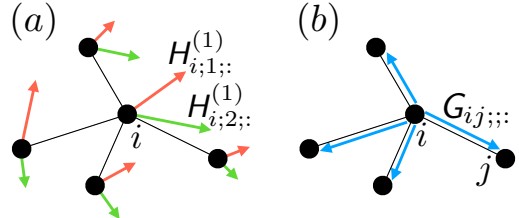

Figure 1: Schematic diagrams of (a) rank-1 tensor field $\mathbf{H}^{(1)}$ with the number of features equaling 2 and (b) the simplest case of $\mathbf{G}_{ij;;:} = \delta_{il}\delta_{jk}A_{ij}\boldsymbol{I}(\boldsymbol{x}_k - \boldsymbol{x}_l) = A_{ij}(\boldsymbol{x}_j - \boldsymbol{x}_i)$.

tation of $H^{(p)}_{i;g;k_1 k_2 \ldots k_p}$ because $i$ and $j$ permutes under the vertex permutation and $k$ represents the spatial index while the number of features is always 1. The IsoAM can be viewed as a weighted adjacency matrix for each direction and reflects spatial information while the usual weighted adjacency matrix cannot because a graph has only one adjacency matrix. If the size of the set $\{G_{ij;;:} \neq \mathbf{0}\}_j$ is greater than or equal to $d$, then it can be deemed to be a frame, which is a generalization of a basis. For the simplest case, one can define $\boldsymbol{T}_{ijkl} = \delta_{il}\delta_{jk}A_{ij}\boldsymbol{I}$ (Figure 1 (b)), where $\delta_{ij}$ is the Kronecker delta, $\boldsymbol{A}$ is the adjacency matrix of the graph, and $\boldsymbol{I}$ is the identity matrix that is the simplest rank-2 tensor. In another case, $\boldsymbol{T}_{ijkl}$ can be determined from the geometry of a graph, as defined in equation 16. Nevertheless, in the bulk of this section, we retain $\boldsymbol{T}_{ijkl}$ abstract to cover various forms of interaction, such as position-aware GNNs (You et al., 2019). Here, $\mathbf{G}$ is composed of only untrainable parameters and thus can be determined before training.

**Proposition 3.1.** *IsoAM defined in equation 4 is both translation invariant and orthogonal transformation equivariant, i.e., for any isometric transformation $\forall \boldsymbol{t} \in \mathbb{R}^3, \boldsymbol{U} \in \mathrm{O}(d), T : \boldsymbol{x} \mapsto \boldsymbol{U}\boldsymbol{x} + \boldsymbol{t}$,*

$$T : \boldsymbol{G}_{ij;;k} \mapsto \sum_l U_{kl} \boldsymbol{G}_{ij;;l}. \tag{5}$$

Based on the definition of the GCN layer in the equation 1, let $\mathbf{G} * \mathbf{H}^{(0)} \in \mathbb{R}^{|\mathcal{V}| \times f \times d}$ denote the **convolution** between $\mathbf{G}$ and the rank-0 tensor field $\mathbf{H}^{(0)} \in \mathbb{R}^{|\mathcal{V}| \times f}$ ($f \in \mathbb{Z}^+$) as follows:

$$(\mathbf{G} * \mathbf{H}^{(0)})_{i;g;k} := \sum_j \mathbf{G}_{ij;;k} H_{j;g;}^{(0)}. \tag{6}$$

With a rank-1 tensor field $\mathbf{H}^{(1)} \in \mathbb{R}^{|\mathcal{V}| \times f \times d}$, let $\mathbf{G} \odot \mathbf{H}^{(1)} \in \mathbb{R}^{|\mathcal{V}| \times f}$ and $\mathbf{G} \odot \mathbf{G} \in \mathbb{R}^{|\mathcal{V}| \times |\mathcal{V}|}$ denote the **contractions** which are defined as follows:

$$(\mathbf{G} \odot \mathbf{H}^{(1)})_{i;g;} := \sum_{j,k} G_{ij;;k} H_{j;g;k}^{(1)}, \quad (\mathbf{G} \odot \mathbf{G})_{il;;} := \sum_{j,k} G_{ij;;k} G_{jl;k}. \tag{7}$$

The contraction of IsoAMs $\mathbf{G} \odot \mathbf{G}$ can be interpreted as the inner product of each component in the IsoAMs. Thus, the subsequent proposition follows.

**Proposition 3.2.** *The contraction of IsoAMs $\mathbf{G} \odot \mathbf{G}$ is isometric transformation invariant, i.e., for any isometric transformation $\forall \boldsymbol{t} \in \mathbb{R}^3, \boldsymbol{U} \in \mathrm{O}(d), T : \boldsymbol{x} \mapsto \boldsymbol{U}\boldsymbol{x} + \boldsymbol{t}, \mathbf{G} \odot \mathbf{G} \mapsto \mathbf{G} \odot \mathbf{G}$.*

With a rank-$p$ tensor field $\mathbf{H}^{(p)} \in \mathbb{R}^{|\mathcal{V}| \times f \times d^p}$, let $\mathbf{G} \otimes \mathbf{H}^{(p)} \in \mathbb{R}^{|\mathcal{V}| \times f \times d^{1+p}}$. and $\mathbf{G} \otimes \mathbf{G} \in \mathbb{R}^{|\mathcal{V}| \times |\mathcal{V}| \times d^2}$ denote the **tensor products** defined as follows:

$$(\mathbf{G} \otimes \mathbf{H}^{(p)})_{i;g;km_1m_2...m_p} := \sum_j \mathbf{G}_{ij;;k} H_{j;g;m_1m_2...m_p}^{(p)}, \tag{8}$$

$$(\mathbf{G} \otimes \mathbf{G})_{il;;k_1k_2} := \sum_j \mathbf{G}_{ij;;k_1} \mathbf{G}_{jl;;k_2}. \tag{9}$$

The tensor product of IsoAMs $\mathbf{G} \otimes \mathbf{G}$ can be interpreted as the tensor product of each of the IsoAMs components. Thus, the subsequent proposition follows:

**Proposition 3.3.** *The tensor product of the IsoAMs $\mathbf{G} \otimes \mathbf{G}$ is isometric transformation equivariant in terms of the rank-2 tensor, i.e., for any isometric transformation $\forall \boldsymbol{t} \in \mathbb{R}^3, \boldsymbol{U} \in \mathrm{O}(d), T : \boldsymbol{x} \mapsto \boldsymbol{U}\boldsymbol{x} + \boldsymbol{t}$, and $\forall i, j \in 1, \ldots, |\mathcal{V}|, (\mathbf{G} \otimes \mathbf{G})_{ij;;k_1k_2} \mapsto \boldsymbol{U}_{k_1l_1} \boldsymbol{U}_{k_2l_2} (\mathbf{G} \otimes \mathbf{G})_{ij;;l_1l_2}$.*

This proposition is easily generalized to the tensors of higher ranks by defining the $p$th tensor power of $\mathbf{G}$ as follows: $\bigotimes^0 \mathbf{G} = 1, \bigotimes^1 \mathbf{G} = \mathbf{G}$, and $\bigotimes^p \mathbf{G} = \bigotimes^{p-1} \mathbf{G} \otimes \mathbf{G}$. Namely, $\bigotimes^p \mathbf{G}$ is isometric transformation equivariant in terms of rank-$p$ tensor. Therefore, one can see that $(\bigotimes^p \mathbf{G}) \otimes \mathbf{H}^{(q)} = (\bigotimes^{p-1} \mathbf{G}) \otimes (\mathbf{G} \otimes \mathbf{H}^{(q)})$. Moreover, the convolution can be generalized for $\bigotimes^p \mathbf{G}$ and the rank-0 tensor field $\mathbf{H}^{(0)} \in \mathbb{R}^{|\mathcal{V}| \times f}$ as follows:

$$\left[\left(\bigotimes^p \mathbf{G}\right) * \mathbf{H}^{(0)}\right]_{i;g;k_1k_2...k_p} = \sum_j \left(\bigotimes^p \mathbf{G}\right)_{ij;;k_1k_2...k_p} H_{j;g;}^{(0)}. \tag{10}$$

The contraction can be generalized for $\bigotimes^p \mathbf{G}$ and the rank-$q$ tensor field $\mathbf{H}^{(q)} \in \mathbb{R}^{|\mathcal{V}| \times f \times d^q}$ ($p \geq q$) as specified below:

$$\left[\left(\bigotimes^p \mathbf{G}\right) \odot \mathbf{H}^{(q)}\right]_{i;g;k_1k_2...k_{p-q}} = \sum_{j,m_1,m_2,...,m_q} \left(\bigotimes^p \mathbf{G}\right)_{ij;;k_1k_2...k_{p-q}m_1m_2...m_q} H_{j;g;m_1m_2...m_q}^{(q)}. \tag{11}$$

For the case $p < q$, the contraction can be defined similarly.

## 3.2 CONSTRUCTION OF ISOGCN

Using the operations defined above, we can construct IsoGCN layers, which take the tensor field of any rank as input, and output the tensor field of any rank, which can differ from those of the input. In addition, one can show that these layers are also equivariant under the vertex permutation, as discussed in Maron et al. (2018).

### 3.2.1 ISOMETRIC TRANSFORMATION INVARIANT LAYER

As can be seen in Proposition 3.1, the contraction of IsoAMs is isometric transformation invariant. Therefore, for the isometric transformation invariant layer with a rank-0 input tensor field $f : \mathbb{R}^{|\mathcal{V}| \times f_{\text{in}}} \ni \boldsymbol{H}_{\text{in}}^{(0)} \mapsto \boldsymbol{H}_{\text{out}}^{(0)} \in \mathbb{R}^{|\mathcal{V}| \times f_{\text{out}}}$ ($f_{\text{in}}, f_{\text{out}} \in \mathbb{Z}^+$), the activation function $\sigma$, and the trainable parameter matrix $\boldsymbol{W} \in \mathbb{R}^{f_{\text{in}} \times f_{\text{out}}}$ can be constructed as $\boldsymbol{H}_{\text{out}}^{(0)} = \sigma\left((\mathbf{G} \odot \mathbf{G}) \boldsymbol{H}_{\text{in}}^{(0)} \boldsymbol{W}\right)$. By defining $\boldsymbol{L} := \mathbf{G} \odot \mathbf{G} \in \mathbb{R}^{|\mathcal{V}| \times |\mathcal{V}|}$, it can be simplified as $\boldsymbol{H}_{\text{out}}^{(0)} = \sigma\left(\boldsymbol{L} \boldsymbol{H}_{\text{in}}^{(0)} \boldsymbol{W}\right)$, which has the same form as a GCN (equation 1), with the exception that $\hat{\boldsymbol{A}}$ is replaced with $\boldsymbol{L}$.

An isometric transformation invariant layer with the rank-$p$ input tensor field $\mathbf{H}_{\text{in}}^{(p)} \in \mathbb{R}^{|\mathcal{V}| \times f_{\text{in}} \times d^p}$ can be formulated as $\boldsymbol{H}_{\text{out}}^{(0)} = F_{p \to 0}(\mathbf{H}_{\text{in}}^{(p)}) = \sigma\left(\left[\bigotimes^p \mathbf{G} \odot \mathbf{H}_{\text{in}}^{(p)}\right] \boldsymbol{W}\right)$. If $p = 1$, such approaches utilize the inner products of the vectors in $\mathbb{R}^d$, these operations correspond to the extractions of a relative distance and an angle of each pair of vertices, which are employed in Klicpera et al. (2020).

### 3.2.2 ISOMETRIC TRANSFORMATION EQUIVARIANT LAYER

To construct an isometric transformation equivariant layer, one can use linear transformation, convolution and tensor product to the input tensors. If both the input and the output tensor ranks are greater than 0, one can apply neither nonlinear activation nor bias addition because these operations will cause an inappropriate distortion of the isometry because isometric transformation does not commute with them in general. However, a conversion that uses only a linear transformation, convolution, and tensor product does not have nonlinearity, which limits the predictive performance of the model. To add nonlinearity to such a conversion, we can first convert the input tensors to rank-0 ones, apply nonlinear activations, and then multiply them to the higher rank tensors.

The nonlinear isometric transformation equivariant layer with the rank-$m$ input tensor field $\mathbf{H}_{\text{in}}^{(m)} \in \mathbb{R}^{|\mathcal{V}| \times f_{\text{in}} \times d^m}$ and the rank-$l$ ($m \leq l$) output tensor $\mathbf{H}_{\text{out}}^{(l)} \in \mathbb{R}^{|\mathcal{V}| \times f_{\text{out}} \times d^l}$ can be defined as:

$$\mathbf{H}_{\text{out}}^{(l)} = F_{m \to 0}\left(\mathbf{H}_{\text{in}}^{(m)}\right) \times \mathbf{F}_{m \to l}\left(\mathbf{H}_{\text{in}}^{(m)}\right), \quad \mathbf{F}_{m \to l}\left(\mathbf{H}_{\text{in}}^{(m)}\right) = \left[\bigotimes^{l-m} \mathbf{G}\right] \otimes \mathbf{H}_{\text{in}}^{(m)} \boldsymbol{W}^{ml}, \quad (12)$$

where $\times$ denotes multiplication with broadcasting and $\boldsymbol{W}^{ml} \in \mathbb{R}^{f_{\text{in}} \times f_{\text{out}}}$ are trainable weight matrices multiplied in the feature direction. If $m = 0$, we regard $\mathbf{G} \otimes \mathbf{H}^{(0)}$ as $\mathbf{G} * \mathbf{H}^{(0)}$. If $m = l$, one can add the residual connection (He et al., 2016) in equation 12. If $m > l$,

$$\mathbf{H}_{\text{out}}^{(l)} = F_{m \to 0}\left(\mathbf{H}_{\text{in}}^{(m)}\right) \times \mathbf{F}_{m \to l}\left(\mathbf{H}_{\text{in}}^{(m)}\right), \quad \mathbf{F}_{m \to l}\left(\mathbf{H}_{\text{in}}^{(m)}\right) = \left[\bigotimes^{m-l} \mathbf{G}\right] \odot \mathbf{H}_{\text{in}}^{(m)} \boldsymbol{W}^{ml}. \quad (13)$$

In general, the nonlinear isometric transformation equivariant layer with the rank-0 to rank-$M$ input tensor field $\{\mathbf{H}_{\text{in}}^{(m)}\}_{m=0}^M$ and the rank-$l$ output tensor field $\mathbf{H}_{\text{out}}^{(l)}$ can be defined as:

$$\mathbf{H}_{\text{out}}^{(l)} = \mathbf{H}_{\text{in}}^{(l)} \boldsymbol{W} + \sum_{m=0}^M f_{\text{gather}}\left(\left\{F_{k \to 0}(\mathbf{H}_{\text{in}}^{(k)})\right\}_{k=0}^M\right) \times \mathbf{F}_{m \to l}\left(\mathbf{H}_{\text{in}}^{(m)}\right), \quad (14)$$

where $f_{\text{gather}}$ denotes a function such as summation, product and concatenation in the feature direction. One can see that this layer is similar to that in the TFN (equation 2), while there are no spherical harmonics and trainable message passing.

To be exact, the output of the layer defined above is translation invariant. To output translation equivariant variables such as the vertex positions after deformation (which change accordingly with the translation of the input graph), one can first define the reference vertex position $\boldsymbol{x}_{\text{ref}}$ for each graph, then compute the translation invariant output using equation 14, and finally, add $\boldsymbol{x}_{\text{ref}}$ to the output. For more detailed information on IsoGCN modeling, see Appendix D.

### 3.3 EXAMPLE OF ISOAM

The IsoGCN $\mathbf{G}$ is defined in a general form for the propositions to work with various classes of graph. In this section, we concretize the concept of the IsoAM to apply an IsoGCN

to mesh structured data. Here, a mesh is regarded as a graph regarding the points in the mesh as vertices of the graph and assuming two vertices are connected when they share the same cell. A concrete instance of IsoAM $\tilde{\mathbf{D}}, \mathbf{D} \in \mathbb{R}^{|\mathcal{V}| \times |\mathcal{V}| \times d}$ is defined as follows:

$$\tilde{D}_{ij;;k} = D_{ij;;k} - \delta_{ij} \sum_l D_{il;;k}, \tag{15}$$

$$D_{ij;;} = \boldsymbol{M}_i^{-1} \frac{\boldsymbol{x}_j - \boldsymbol{x}_i}{\|\boldsymbol{x}_j - \boldsymbol{x}_i\|^2} w_{ij} A_{ij}(m), \tag{16}$$

$$\boldsymbol{M}_i = \sum_l \frac{\boldsymbol{x}_l - \boldsymbol{x}_i}{\|\boldsymbol{x}_l - \boldsymbol{x}_i\|} \otimes \frac{\boldsymbol{x}_l - \boldsymbol{x}_i}{\|\boldsymbol{x}_l - \boldsymbol{x}_i\|} w_{il} A_{il}(m), \tag{17}$$

where $\mathbb{R}^{|\mathcal{V}| \times |\mathcal{V}|} \ni \boldsymbol{A}(m) := \min\left(\sum_{k=1}^m \boldsymbol{A}^k, 1\right)$ is an adjacency matrix up to $m$ hops and $w_{ij} \in \mathbb{R}$ is an untrainable weight between the $i$th and $j$th vertices that is determined depending on the tasks[2]. By regarding

Table 1: Correspondence between the differential operators and the expressions using the IsoAM $\tilde{\mathbf{D}}$.

| Differential op. | Expression |
| --- | --- |
| Gradient | $\tilde{\mathbf{D}} * \mathbf{H}^{(0)}$ |
| Divergence | $\tilde{\mathbf{D}} \odot \mathbf{H}^{(1)}$ |
| Laplacian | $\tilde{\mathbf{D}} \odot \tilde{\mathbf{D}} \mathbf{H}^{(0)}$ |
| Jacobian | $\tilde{\mathbf{D}} \otimes \mathbf{H}^{(1)}$ |
| Hessian | $\tilde{\mathbf{D}} \otimes \tilde{\mathbf{D}} * \mathbf{H}^{(0)}$ |

$\boldsymbol{T}_{ijkl} = \delta_{il}\delta_{jk}\boldsymbol{M}_i^{-1}w_{ij}\boldsymbol{A}_{ij}(m)/\|\boldsymbol{x}_j - \boldsymbol{x}_i\|^2$ in equation 4, one can see that $\mathbf{D}$ is qualified as an IsoAM. Because a linear combination of IsoAMs is also an IsoAM, $\tilde{\mathbf{D}}$ is an IsoAM. Thus, they provide both translation invariance and orthogonal transformation equivariance. $\tilde{\mathbf{D}}$ can be obtained only from the mesh geometry information, thus can be computed in the preprocessing step.

Here, $\tilde{\mathbf{D}}$ is designed such that it corresponds to the gradient operator model used in physical simulations (Tamai & Koshizuka, 2014; Swartz & Wendroff, 1969). As presented in Table 1 and Appendix C, $\tilde{\mathbf{D}}$ is closely related to many differential operators, such as the gradient, divergence, Laplacian, Jacobian, and Hessian. Therefore, the considered IsoAM plays an essential role in constructing neural network models that are capable of learning differential equations.

## 4 EXPERIMENT

To test the applicability of the proposed model, we composed the following two datasets: 1) a differential operator dataset of grid meshes; and 2) an anisotropic nonlinear heat equation dataset of meshes generated from CAD data. In this section, we discuss our machine learning model, the definition of the problem, and the results for each dataset.

Using $\tilde{\mathbf{D}}$ defined in Section 3.3, we constructed a neural network model considering an encode-process-decode configuration (Battaglia et al., 2018). The encoder and decoder were comprised of component-wise MLPs and tensor operations. For each task, we tested $m = 2, 5$ in equation 16 to investigate the effect of the number of hops considered. In addition to the GCN (Kipf & Welling, 2017), we chose GIN (Xu et al., 2018), SGCN (Wu et al., 2019), Cluster-GCN (Chiang et al., 2019), and GCNII (Chen et al., 2020) as GCN variant baseline models. For the equivariant models, we chose the TFN (Thomas et al., 2018) and SE(3)-Transformer (Fuchs et al., 2020) as the baseline. We implemented these models using PyTorch 1.6.0 (Paszke et al., 2019) and PyTorch Geometric 1.6.1 (Fey & Lenssen, 2019). For both the TFN and SE(3)-Transformer, we used implementation of Fuchs et al. (2020) [3] because the computation of the TFN is considerably faster than the original implementation, as claimed in Fuchs et al. (2020). For each experiment, we minimized the mean squared loss using the Adam optimizer (Kingma & Ba, 2014). The corresponding implementation and the dataset will be made available online. The details of the experiments can be found in Appendix E and F.

### 4.1 DIFFERENTIAL OPERATOR DATASET

To demonstrate the expressive power of IsoGCNs, we created a dataset to learn the differential operators. We first generated a pseudo-2D grid mesh randomly with only one cell in the $Z$ direction and 10 to 100 cells in the $X$ and $Y$ directions. We then generated scalar fields on the grid meshes

---

[2]$\boldsymbol{M}_i$ is invertible when the number of independent vectors in $\{\boldsymbol{x}_l - \boldsymbol{x}_i\}_l$ is greater than or equal to the space dimension $d$, which is true for common meshes, e.g., a solid mesh in 3D Euclidean space.

[3]https://github.com/FabianFuchsML/se3-transformer-public

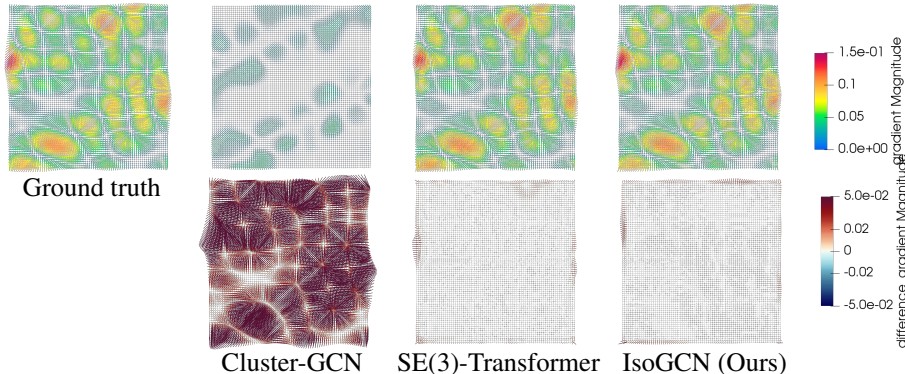

Ground truth

Cluster-GCN     SE(3)-Transformer     IsoGCN (Ours)

Figure 2: (Top) the gradient field and (bottom) the error vector between the prediction and the ground truth of a test data sample. The error vectors are exaggerated by a factor of 2 for clear visualization.

Table 2: Summary of the test losses (mean squared error $\pm$ the standard error of the mean in the original scale) of the differential operator dataset: $0 \rightarrow 1$ (the scalar field to the gradient field), $0 \rightarrow 2$ (the scalar field to the Hessian field), $1 \rightarrow 0$ (the gradient field to the Laplacian field), and $1 \rightarrow 2$ (the gradient field to the Hessian field). Here, if "$\boldsymbol{x}$" is "Yes", $\boldsymbol{x}$ is also in the input feature. We show only the best setting for each method except for the equivariant models. For a full table, see Appendix E. OOM denotes the out-of-memory on the applied GPU (32 GiB).

| Method | # hops | $\boldsymbol{x}$ | Loss of $0 \rightarrow 1$ $\times 10^{-5}$ | Loss of $0 \rightarrow 2$ $\times 10^{-6}$ | Loss of $1 \rightarrow 0$ $\times 10^{-6}$ | Loss of $1 \rightarrow 2$ $\times 10^{-6}$ |
|---|---|---|---|---|---|---|
| GIN | 5 | Yes | $147.07 \pm 0.51$ | $47.35 \pm 0.35$ | $404.92 \pm 1.74$ | $46.18 \pm 0.39$ |
| GCNII | 5 | Yes | $151.13 \pm 0.53$ | $31.87 \pm 0.22$ | $280.61 \pm 1.30$ | $39.38 \pm 0.34$ |
| SGCN | 5 | Yes | $151.16 \pm 0.53$ | $55.08 \pm 0.42$ | $127.21 \pm 0.63$ | $56.97 \pm 0.44$ |
| GCN | 5 | Yes | $151.14 \pm 0.53$ | $48.50 \pm 0.35$ | $542.30 \pm 2.14$ | $25.37 \pm 0.28$ |
| Cluster-GCN | 5 | Yes | $146.91 \pm 0.51$ | $26.60 \pm 0.19$ | $185.21 \pm 0.99$ | $18.18 \pm 0.20$ |
| TFN | 2 | No | $2.47 \pm 0.02$ | OOM | $26.69 \pm 0.24$ | OOM |
|  | 5 | No | OOM | OOM | OOM | OOM |
| SE(3)-Trans. | 2 | No | $\mathbf{1.79} \pm 0.02$ | $\mathbf{3.50} \pm 0.04$ | $\mathbf{2.52} \pm 0.02$ | OOM |
|  | 5 | No | $2.12 \pm 0.02$ | OOM | $7.66 \pm 0.05$ | OOM |
| **IsoGCN** (Ours) | 2 | No | $2.67 \pm 0.02$ | $6.37 \pm 0.07$ | $7.18 \pm 0.06$ | $\mathbf{1.44} \pm 0.02$ |
|  | 5 | No | $14.19 \pm 0.10$ | $21.72 \pm 0.25$ | $34.09 \pm 0.19$ | $8.32 \pm 0.09$ |

and analytically calculated the gradient, Laplacian, and Hessian fields. We generated 100 samples for each train, validation, and test dataset. For simplicity, we set $w_{ij} = 1$ in equation 16 for all $(i, j) \in \mathcal{E}$. To compare the performance with the GCN models, we simply replaced an IsoGCN layer with a GCN or its variant layers while keeping the number of hops $m$ the same to enable a fair comparison. We adjusted the hyperparameters for the equivariant models to ensure that the number of parameters in each was almost the same as that in the IsoGCN model. For more details regarding the model architecture, see Appendix E. We conducted the experiments using the following settings: 1) inputting the scalar field and predicting the gradient field (rank-0 $\rightarrow$ rank-1 tensor); 2) inputting the scalar field and predicting the Hessian field (rank-0 $\rightarrow$ rank-2 tensor); 3) inputting the gradient field and predicting the Laplacian field (rank-1 $\rightarrow$ rank-0 tensor); and 4) inputting the gradient field and predicting the Hessian field (rank-1 $\rightarrow$ rank-2 tensor).

Figure 2 and Table 2 present a visualization and comparison of predictive performance, respectively. The results show that an IsoGCN outperforms other GCN models for all settings. This is because the IsoGCN model has information on the relative position of the adjacency vertices, and thus understands the direction of the gradient, whereas the other GCN models cannot distinguish where the adjacencies are, making it nearly impossible to predict the gradient directions. Adding the vertex positions to the input feature to other GCN models exhibited a performance improvement, however as the vertex position is not a translation invariant feature, it could degrade the predictive performance of the models. Thus, we did not input $\boldsymbol{x}$ as a vertex feature to the IsoGCN model or other equivariant models to retain their isometric transformation invariant and equivariant natures. IsoGCNs perform competitively against other equivariant models with shorter inference time as shown in Ta-

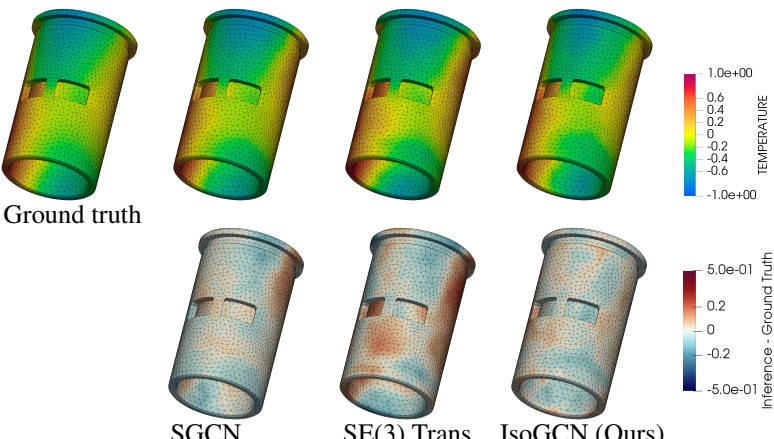

Figure 3: (Top) the temperature field of the ground truth and inference results and (bottom) the error between the prediction and the ground truth of a test data sample. The error is exaggerated by a factor of 2 for clear visualization.

ble 7. As mentioned in Section 3.3, $\tilde{\mathbf{D}}$ corresponds to the gradient operator, which is now confirmed in practice.

### 4.2 ANISOTROPIC NONLINEAR HEAT EQUATION DATASET

To apply the proposed model to a real problem, we adopted the anisotropic nonlinear heat equation. We considered the task of predicting the time evolution of the temperature field based on the initial temperature field, material property, and mesh geometry information as inputs. We randomly selected 82 CAD shapes from the first 200 shapes of the ABC dataset (Koch et al., 2019), generate first-order tetrahedral meshes using a mesh generator program, Gmsh (Geuzaine & Remacle, 2009), randomly set the initial temperature and anisotropic thermal conductivity, and finally conducted a finite element analysis (FEA) using the FEA program FrontISTR[4] (Morita et al., 2016; Ihara et al., 2017).

For this task, we set $w_{ij} = V_j^{\text{effective}}/V_i^{\text{effective}}$, where $V_i^{\text{effective}}$ denotes the effective volume of the $i$th vertex (equation 46.) Similarly to the differential operator dataset, we tested the number of hops $m = 2, 5$. However because we put four IsoAM operations in one model, the number of hops visible from the model is 8 ($m = 2$) or 20 ($m = 5$). As is the case with the differential operator dataset, we replaced an IsoGCN layer accordingly for GCN or its variant models. In the case of $k = 2$, we reduced the number of parameters for each of the equivariant models to fewer than the IsoGCN model because they exceeded the memory of the GPU (NVIDIA Tesla V100 with 32 GiB memory) with the same number of parameters. In the case of $k = 5$, neither the TFN nor the SE(3)-Transformer fits into the memory of the GPU even with the number of parameters equal to 10. For more details about the dataset and the model, see Appendix F.

Figure 3 and Table 3 present the results of the qualitative and quantitative comparisons for the test dataset. The IsoGCN demonstrably outperforms all other baseline models. Moreover, owing to the computationally efficient isometric transformation invariant nature of IsoGCNs, it also achieved a high prediction performance for the meshes that had a significantly larger graph than those considered in the training dataset. The IsoGCN can scale up to 1M vertices, which is practical and is considerably greater than that reported in Sanchez-Gonzalez et al. (2020). Therefore, we conclude that IsoGCN models can be trained on relatively smaller meshes[5] to save the training time and then used to apply the inference to larger meshes without observing significant performance deterioration.

Table 4 reports the preprocessing and inference computation time using the equivariant models with $m = 2$ as the number of hops and FEA using FrontISTR 5.0.0. We varied the time step ($\Delta t = $

---

[4]`https://github.com/FrontISTR/FrontISTR`. We applied a private update to FrontISTR to deal with the anisotropic heat problem, which will be also made available online.

[5]However, it should also be sufficiently large to express sample shapes and fields.

1.0, 0.5) for the FEA computation to compute the $t = 1.0$ time evolution thus, resulting in different computation times and errors compared to an FEA with $\Delta t = 0.01$, which was considered as the ground truth. Clearly, the IsoGCN is 3- to 5- times faster than the FEA with the same level of accuracy, while other equivariant models have almost the same speed as FrontISTR with $\Delta t = 0.5$.

## 5 CONCLUSION

In this study, we proposed the GCN-based isometric transformation invariant and equivariant models called *IsoGCN*. We discussed an example of an isometric adjacency matrix (IsoAM) that was closely related to the essential differential operators. The experiment results confirmed that the proposed model leveraged the spatial structures and can deal with large-scale graphs. The computation time of the IsoGCN model is significantly shorter than the FEA, which other equivariant models cannot achieve. Therefore, IsoGCN must be the first choice to learn physical simulations because of its computational efficiency as well as isometric transformation invariance and equivariance. Our demonstrations were conducted on the mesh structured dataset based on the FEA results. However, we expect IsoGCNs to be applied to various domains, such as object detection, molecular property prediction, and physical simulations using particles.

Table 3: Summary of the test losses (mean squared error $\pm$ the standard error of the mean in the original scale) of the anisotropic nonlinear heat dataset. Here, if "$\boldsymbol{x}$" is "Yes", $\boldsymbol{x}$ is also in the input feature. We show only the best setting for each method except for the equivariant models. For the full table, see Appendix E. OOM denotes the out-of-memory on the applied GPU (32 GiB).

| Method | # hops | $x$ | Loss $\times 10^{-3}$ |
|---|---|---|---|
| GIN | 2 | No | $16.921 \pm 0.040$ |
| GCN | 2 | No | $10.427 \pm 0.028$ |
| GCNII | 5 | No | $8.377 \pm 0.024$ |
| Gluster-GCN | 2 | No | $7.266 \pm 0.021$ |
| SGCN | 5 | No | $6.426 \pm 0.018$ |
| TFN | 2 | No | $15.661 \pm 0.019$ |
| | 5 | No | OOM |
| SE(3)-Trans. | 2 | No | $14.164 \pm 0.018$ |
| | 5 | No | OOM |
| **IsoGCN** (Ours) | 2 | No | $4.674 \pm 0.014$ |
| | 5 | No | $\mathbf{2.470} \pm 0.008$ |

ACKNOWLEDGMENTS

The authors gratefully acknowledge Takanori Maehara for his helpful advice and NVIDIA for hardware donations. This work was supported by JSPS KAKENHI Grant Number 19H01098.

Table 4: Comparison of computation time. To generate the test data, we sampled CAD data from the test dataset and then generated the mesh for the graph to expand while retaining the element volume at almost the same size. The initial temperature field and the material properties are set randomly using the same methodology as the dataset sample generation. For a fair comparison, each computation was run on the same CPU (Intel Xeon E5-2695 v2@2.40GHz) using one core, and we excluded file I/O time from the measured time. OOM denotes the out-of-memory (500 GiB).

| Method | $|\mathcal{V}| = 21,289$ Loss $\times 10^{-4}$ | Time [s] | $|\mathcal{V}| = 155,019$ Loss $\times 10^{-4}$ | Time [s] | $|\mathcal{V}| = 1,011,301$ Loss $\times 10^{-4}$ | Time [s] |
|---|---|---|---|---|---|---|
| FrontISTR ($\Delta t = 1.0$) | 10.9 | 16.7 | 6.1 | 181.7 | 2.9 | 1656.5 |
| FrontISTR ($\Delta t = 0.5$) | 0.8 | 30.5 | 0.4 | 288.0 | 0.2 | 2884.2 |
| TFN | 77.9 | 46.1 | 30.1 | 400.9 | OOM | OOM |
| SE(3)-Transformer | 111.4 | 31.2 | 80.3 | 271.1 | OOM | OOM |
| **IsoGCN** (Ours) | 8.1 | **7.4** | 4.9 | **84.1** | 3.9 | **648.4** |

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

## A  NOTATION

| | |
|---|---|
| $\mathcal{G}$ | A graph |
| $\mathcal{V}$ | A vertex set |
| $\|\mathcal{V}\|$ | The number of vertices |
| $\mathcal{E}$ | An edge set |
| $\mathbb{Z}^+$ | The positive integers |
| $d$ | The dimension of the Euclidean space |
| $\boldsymbol{x}_i$ | The position of the $i$th vertex |
| $x_{ik}$ | Element $k$ of $\boldsymbol{x}_i$ |
| $\mathbf{G} \in \mathbb{R}^{\|\mathcal{V}\| \times \|\mathcal{V}\| \times d}$ | The isometric adjacency matrix (IsoAM) (equation 4) |
| $\mathbf{G}_{ij;:} \in \mathbb{R}^d$ | Slice of $\mathbf{G}$ in the spatial index (equation 4) |
| $\mathbf{G}_{ij;:k} \in \mathbb{R}$ | Element $(i, j, k)$ of $\mathbf{G}$ |
| $\mathbf{H}^{(p)} \in \mathbb{R}^{\|\mathcal{V}\| \times f \times d^p}$ | A rank-$p$ tensor field tensor ($f, p \in \mathbb{Z}^+$) |
| $H^{(p)}_{i;g;k_1 k_2 \ldots k_p}$ | Element $(i; g; k_1, k_2, \ldots, k_p)$ of $\mathbf{H}^{(p)}$. $i$ refers to the permutation representation, $k_1, \ldots k_p$ refer to the Euclidean representation, and $g$ denotes the feature index (See section 3). |
| $\left[ \bigotimes^{p} \mathbf{G} \right] * \mathbf{H}^{(0)}$ | Convolution of the $p$th power of $\mathbf{G}$ and rank-0 tensor field $\mathbf{H}^{(0)}$ (equation 6, equation 10) |
| $\left[ \bigotimes^{p} \mathbf{G} \right] \odot \mathbf{H}^{(q)}$ | Contraction of the $p$th power of $\mathbf{G}$ and rank-$q$ tensor fields (equation 7, equation 11) |
| $\left[ \bigotimes^{p} \mathbf{G} \right] \otimes \mathbf{H}^{(q)}$ | Tensor product of the $p$th power of $\mathbf{G}$ and rank-$q$ tensor fields $\mathbf{H}^{(q)}$ (equation 8) |
| $\mathbf{H}^{(p)}_{\text{in}}$ | The rank-$p$ input tensor field of the considered layer |
| $\mathbf{H}^{(p)}_{\text{out}}$ | The rank-$p$ output tensor field of the considered layer |
| $\sigma$ | The activation function |
| $\boldsymbol{W}$ | The trainable parameter matrix |
| $\boldsymbol{A} \in \mathbb{R}^{\|\mathcal{V}\| \times \|\mathcal{V}\|}$ | An adjacency matrix |
| $\delta_{ij}$ | The Kronecker delta |
| $V^{\text{effective}}_i$ | The effective volume of the $i$th vertex (equation 46) |
| $V^{\text{mean}}_i$ | The mean volume of the $i$th vertex (equation 47) |
| $\tilde{\mathbf{D}} \in \mathbb{R}^{\|\mathcal{V}\| \times \|\mathcal{V}\|}$ | A concrete instance of IsoAM (equation 15) |

## B  PROOFS OF PROPOSITIONS

In this section, we present the proofs of the propositions described in Section 3. Let $\mathbb{R}^3 \ni g(\boldsymbol{x}_l, \boldsymbol{x}_k) = (\boldsymbol{x}_k - \boldsymbol{x}_l)$. Note that $\mathbf{G}$ is expressed using $g(\boldsymbol{x}_i, \boldsymbol{x}_j)$ as $\mathbf{G}_{ij;;;} = \sum_{k,l \in \mathcal{V}, k \neq l} \boldsymbol{T}_{ijkl} g(\boldsymbol{x}_l, \boldsymbol{x}_k)$.

### B.1  PROOF OF PROPOSITION 3.1

*Proof.* First, we demonstrate the invariance with respect to the translation with $\forall \boldsymbol{t} \in \mathbb{R}^d$. $g(\boldsymbol{x}_i, \boldsymbol{x}_j)$ is transformed invariantly as follows under translation:

$$
\begin{aligned}
g(\boldsymbol{x}_i + \boldsymbol{t}, \boldsymbol{x}_j + \boldsymbol{t}) &= [\boldsymbol{x}_j + \boldsymbol{t} - (\boldsymbol{x}_i + \boldsymbol{t})] \\
&= (\boldsymbol{x}_j - \boldsymbol{x}_i) \\
&= g(\boldsymbol{x}_i, \boldsymbol{x}_j).
\end{aligned} \tag{18}
$$

By definition, $\boldsymbol{T}_{ijkl}$ is also translation invariant. Thus,

$$
\begin{aligned}
\sum_{k,l \in \mathcal{V}, k \neq l} \boldsymbol{T}_{ijkl} g(\boldsymbol{x}_l + \boldsymbol{t}, \boldsymbol{x}_k + \boldsymbol{t}) &= \sum_{k,l \in \mathcal{V}, k \neq l} \boldsymbol{T}_{ijkl} g(\boldsymbol{x}_l, \boldsymbol{x}_k) \\
&= \mathbf{G}_{ij;;;}.
\end{aligned} \tag{19}
$$

We then show an equivariance regarding the orthogonal transformation with $\forall \boldsymbol{U} \in \mathrm{O}(d)$. $g(\boldsymbol{x}_i, \boldsymbol{x}_j)$ is transformed as follows by orthogonal transformation:

$$
\begin{aligned}
g(\boldsymbol{U}\boldsymbol{x}_i, \boldsymbol{U}\boldsymbol{x}_j) &= \boldsymbol{U}\boldsymbol{x}_j - \boldsymbol{U}\boldsymbol{x}_i \\
&= \boldsymbol{U}(\boldsymbol{x}_j - \boldsymbol{x}_i) \\
&= \boldsymbol{U} g(\boldsymbol{x}_i, \boldsymbol{x}_j).
\end{aligned} \tag{20}
$$

By definition, $\boldsymbol{T}_{ijkl}$ is transformed to $\boldsymbol{U}\boldsymbol{T}_{ijkl}\boldsymbol{U}^{-1}$ by orthogonal transformation. Thus,

$$
\begin{aligned}
\sum_{k,l \in \mathcal{V}, k \neq l} \boldsymbol{U}\boldsymbol{T}_{ijkl}\boldsymbol{U}^{-1} g(\boldsymbol{U}\boldsymbol{x}_l, \boldsymbol{U}\boldsymbol{x}_k) &= \sum_{k,l \in \mathcal{V}, k \neq l} \boldsymbol{U}\boldsymbol{T}_{ijkl}\boldsymbol{U}^{-1}\boldsymbol{U} g(\boldsymbol{x}_l, \boldsymbol{x}_k) \\
&= \boldsymbol{U}\mathbf{G}_{ij;;;}.
\end{aligned} \tag{21}
$$

Therefore, $\mathbf{G}$ is both translation invariant and an orthogonal transformation equivariant. $\square$

### B.2  PROOF OF PROPOSITION 3.2

*Proof.* Here, $\mathbf{G} \odot \mathbf{G}$ is translation invariant because $\mathbf{G}$ is translation invariant. We prove rotation invariance under an orthogonal transformation $\forall \boldsymbol{U} \in \mathrm{O}(n)$. In addition, $\mathbf{G} \odot \mathbf{G}$ is transformed under $\boldsymbol{U}$ as follows:

$$
\begin{aligned}
\sum_{j,k} G_{ij;;k} G_{jl;;k} &\mapsto \sum_{j,k,m,n} U_{km} G_{ij;;m} U_{kn} G_{jl;;n} \\
&= \sum_{j,k,m,n} U_{km} U_{kn} G_{ij;;m} G_{jl;;n} \\
&= \sum_{j,k,m,n} U_{mk}^T U_{kn} G_{ij;;m} G_{jl;;n} \\
&= \sum_{j,m,n} \delta_{mn} G_{ij;;m} G_{jl;;n} && (\because \text{property of the orthogonal matrix}) \\
&= \sum_{j} G_{ij;;m} G_{jl;;m} \\
&= \sum_{j,k} G_{ij;;k} G_{jl;;k}. && (\because \text{Change the dummy index } m \to k) \quad (22)
\end{aligned}
$$

Therefore, $\mathbf{G} \odot \mathbf{G}$ is isometric transformation invariant. $\square$

## B.3 PROOF OF PROPOSITION 3.3

*Proof.* $\mathbf{G} \otimes \mathbf{G}$ is transformed under $\forall \boldsymbol{U} \in \mathrm{O}(n)$ as follows:

$$
\begin{aligned}
\sum_j \boldsymbol{G}_{ij;;k}\boldsymbol{G}_{jl;;m} &\mapsto \sum_{n,o} \boldsymbol{U}_{kn}\boldsymbol{G}_{ij;;n}\boldsymbol{U}_{mo}\boldsymbol{G}_{jl;;o} \\
&= \sum_{n,o} \boldsymbol{U}_{kn}\boldsymbol{G}_{ij;;n}\boldsymbol{G}_{jl;;o}\boldsymbol{U}_{om}^T.
\end{aligned} \tag{23}
$$

By regarding $\boldsymbol{G}_{ij;;n}\boldsymbol{G}_{jl;;o}$ as one matrix $H_{no}$, it follows the coordinate transformation of rank-2 tensor $\boldsymbol{U}\boldsymbol{H}\boldsymbol{U}^T$ for each $i$, $j$, and $l$. □

## C PHYSICAL INTUITION OF $\tilde{\mathbf{D}}$

In this section, we discuss the connection between the concrete IsoAM example $\tilde{\mathbf{D}}$ and the differential operators such as the gradient, divergence, the Laplacian, the Jacobian, and the Hessian operators.

Let $\phi_i \in \mathbb{R}$ denote a rank-0 tensor (scalar) at the $i$th vertex. Let us assume a partial derivative model of a rank-0 tensor $\phi$ at the $i$th vertex regarding the $k$th axis $(\partial\phi/\partial x_k)_i \in \mathbb{R}$ ($k \in \{1, \ldots, d\}$), that is based on the gradient model in the least squares moving particle semi-implicit method (Tamai & Koshizuka, 2014).

$$
\left(\frac{\partial\phi}{\partial x_k}\right)_i := \boldsymbol{M}_i^{-1} \sum_j \frac{\phi_j - \phi_i}{\|\boldsymbol{x}_j - \boldsymbol{x}_i\|} \frac{x_{jk} - x_{ik}}{\|\boldsymbol{x}_j - \boldsymbol{x}_i\|} w_{ij} A_{ij}(m) \tag{24}
$$

$$
= \sum_j \boldsymbol{D}_{ijk}(\phi_j - \phi_i), \tag{25}
$$

$$
\boldsymbol{M}_i = \sum_l \frac{\boldsymbol{x}_l - \boldsymbol{x}_i}{\|\boldsymbol{x}_l - \boldsymbol{x}_i\|} \otimes \frac{\boldsymbol{x}_l - \boldsymbol{x}_i}{\|\boldsymbol{x}_l - \boldsymbol{x}_i\|} w_{il} A_{il}(m). \tag{26}
$$

Although one could define $w_{ij}$ as a function of the distance $\|\boldsymbol{x}_j - \boldsymbol{x}_i\|$, $w_{ij}$ was kept constant with respect to the distance required to maintain the simplicity of the model with fewer hyperparameters.

### C.1 GRADIENT

$\tilde{\mathbf{D}}$ can be viewed as a Laplacian matrix based on $\mathbf{D}$; however, $\tilde{\mathbf{D}} * \mathbf{H}^{(0)}$ can be interpreted as the gradient within the Euclidean space. Let $\nabla \mathbf{H}^{(0)} \in \mathbb{R}^{|\mathcal{V}| \times f \times d}$ be an approximation of the gradient of $\mathbf{H}^{(0)}$. Using equation 25, the gradient model can be expressed as follows:

$$
\left(\nabla \mathbf{H}^{(0)}\right)_{i;g;k} = \frac{\partial H_{i;g;}^{(0)}}{\partial x_k} \tag{27}
$$

$$
= D_{ijk}(H_{j;g;}^{(0)} - H_{i;g;}^{(0)}). \tag{28}
$$

Using this gradient model, we can confirm that $(\tilde{\mathbf{D}} * \mathbf{H}^{(0)})_{i;g;k} = (\nabla \mathbf{H}^{(0)})_{i;glk}$ because

$$
\begin{aligned}
\left(\tilde{\mathbf{D}} * \mathbf{H}^{(0)}\right)_{i;g;} &= \sum_j \tilde{D}_{ij;;k} H_{j;g;}^{(0)} \qquad\qquad (29) \\
&= \sum_j (D_{ij;;k} - \delta_{ij} \sum_l D_{il;;k}) H_{j;g;}^{(0)} \\
&= \sum_j D_{ij;;k} H_{j;g;}^{(0)} - \sum_{j,l} \delta_{ij} D_{il;;k} H_{j;g;}^{(0)} \\
&= \sum_j D_{ij;;k} H_{j;g;}^{(0)} - \sum_l D_{il;;k} H_{i;g;}^{(0)} \\
&= \sum_j D_{ij;;k} H_{j;g;}^{(0)} - \sum_j D_{ij;;k} H_{i;g;}^{(0)} \qquad (\because \text{Change the dummy index } l \to j) \\
&= \sum_j D_{ij;;k} (H_{j;g;}^{(0)} - H_{i;g;}^{(0)}) \\
&= \left(\nabla \mathbf{H}^{(0)}\right)_{i;g;k}. \qquad\qquad (30)
\end{aligned}
$$

Therefore, $\tilde{\mathbf{D}}*$ can be interpreted as the gradient operator within a Euclidean space.

## C.2 DIVERGENCE

We show that $\tilde{\mathbf{D}} \odot \mathbf{H}^{(1)}$ corresponds to the divergence. Using $\mathbf{D}$, the divergence model $\nabla \cdot \mathbf{H}^{(1)} \in \mathbb{R}^{|\mathcal{V}| \times f}$ is expressed as follows:

$$
\begin{aligned}
\left(\nabla \cdot \mathbf{H}^{(1)}\right)_{i;g;} &= \left(\sum_k \frac{\partial \mathbf{H}^{(1)}}{\partial x_k}\right)_{i;g;} \qquad\qquad (31) \\
&= \sum_{j,k} D_{ij;;k} (H_{j;g;k}^{(1)} - H_{i;g;k}^{(1)}). \qquad\qquad (32)
\end{aligned}
$$

Then, $\tilde{\mathbf{D}} \odot \mathbf{H}^{(1)}$ is

$$
\begin{aligned}
(\tilde{\mathbf{D}} \odot \mathbf{H}^{(1)})_{i;g;} &= \sum_{j,k} \tilde{\mathbf{D}}_{ij;;k} H_{i;g;k}^{(1)} \\
&= \sum_{j,k} \left(\mathbf{D}_{ij;;k} - \delta_{ij} \sum_l \mathbf{D}\right) H_{i;g;k}^{(1)} \\
&= \sum_{j,k} \mathbf{D}_{ij;;k} \mathbf{H}_{j;g;k}^{(1)} - \sum_{l,k} \mathbf{D}_{il;;k} \mathbf{H}_{i;g;k}^{(1)} \\
&= \sum_{j,k} D_{ij;;k} (H_{j;g;k}^{(1)} - H_{i;g;k}^{(1)}) \qquad (\because \text{Change the dummy index } l \to j) \\
&= (\nabla \cdot \mathbf{H}^{(1)})_{i;g;}. \qquad\qquad (33)
\end{aligned}
$$

## C.3 LAPLACIAN OPERATOR

We prove that $\tilde{\mathbf{D}} \odot \tilde{\mathbf{D}}$ corresponds to the Laplacian operator within a Euclidean space.

Using equation 25, the Laplacian model $\nabla^2 \mathbf{H}^{(0)} \in \mathbb{R}^{|\mathcal{V}| \times f}$ can be expressed as follows:

$$
\begin{aligned}
\left(\nabla^2 \mathbf{H}^{(0)}\right)_{i;g;} &:= \sum_k \left[ \frac{\partial}{\partial x_k} \left( \frac{\partial \mathbf{H}}{\partial x_k} \right)_i \right]_{i;g;} \\
&= \sum_{j,k} D_{ij;;k} \left[ \left( \frac{\partial \mathbf{H}}{\partial x_k} \right)_{j;g;} - \left( \frac{\partial \mathbf{H}}{\partial x_k} \right)_{i;g;} \right] \\
&= \sum_{j,k} D_{ij;;k} \left[ \sum_l D_{jl;;k}(H^{(0)}_{l;g;} - H^{(0)}_{j;g;}) - \sum_l D_{il;;k}(H^{(0)}_{l;g;} - H^{(0)}_{i;g;}) \right] \\
&= \sum_{j,k,l} D_{ij;;k}(D_{jl;;k} - D_{il;;k})(H^{(0)}_{l;g;} - H^{(0)}_{j;g;}).
\end{aligned}
\tag{34}
$$

Then, $(\tilde{\mathbf{D}} \odot \tilde{\mathbf{D}})\mathbf{H}^{(0)}$ is

$$
\begin{aligned}
((\tilde{\mathbf{D}} \odot \tilde{\mathbf{D}})\mathbf{H}^{(0)})_{i;g;} &= \sum_{j,k,l} \tilde{D}_{ij;;k}\tilde{D}_{jl;;k}H^{(0)}_{l;g;} \\
&= \sum_{j,k,l} \left( D_{ij;;k} - \delta_{ij}\sum_m D_{im;;k} \right) \left( D_{jl;;k} - \delta_{jl}\sum_n D_{jn;;k} \right) H^{(0)}_{l;g;} \\
&= \sum_{j,k,l} D_{ij;;k}D_{jl;;k}H^{(0)}_{l;g;} - \sum_{j,k,n} D_{ij;;k}D_{jn;;k}H^{(0)}_{j;g;} \\
&\quad - \sum_{k,l,m} D_{im;;k}D_{il;;k}H^{(0)}_{l;g;} + \sum_{k,m,n} D_{im;;k}D_{in;;k}H^{(0)}_{i;g;} \\
&= \sum_{j,k,l} D_{ij;;k}D_{jl;;k}H^{(0)}_{l;g;} - \sum_{j,k,n} D_{ij;;k}D_{jn;;k}H^{(0)}_{j;g;} \\
&\quad - \sum_{k,l,j} D_{ij;;k}D_{il;;k}H^{(0)}_{l;g;} + \sum_{k,j,n} D_{ij;;k}D_{in;;k}H^{(0)}_{i;g;} \\
&\quad (\because \text{Change the dummy index } m \to j \text{ for the third and fourth terms}) \\
&= \sum_{j,k,l} D_{ij;;k}(D_{jl;;k} - D_{il;;k})(H^{(0)}_{l;g;} - H^{(0)}_{j;g;}) \\
&\quad (\because \text{Change the dummy index } n \to l \text{ for the second and fourth terms}) \\
&= \left(\nabla^2 \mathbf{H}^{(0)}\right)_{i;g;}.
\end{aligned}
\tag{35}
$$

## C.4 JACOBIAN AND HESSIAN OPERATORS

Considering a similar discussion, we can show the following dependencies. For the Jacobian model, $\mathbf{J}[\mathbf{H}^{(1)}] \in \mathbb{R}^{|\mathcal{V}| \times f \times d \times d}$,

$$
\begin{aligned}
\left(\mathbf{J}[\mathbf{H}^{(1)}]\right)_{i;g;kl} &= \left( \frac{\partial \mathbf{H}^{(1)}}{\partial x_l} \right)_{i;g;k} \tag{36} \\
&= \sum_j D_{ij;;l}(H^{(1)}_{j;g;k} - H^{(1)}_{i;g;k}) \tag{37} \\
&= (\tilde{D} \otimes \mathbf{H}^{(1)})_{i;g;lk}. \tag{38}
\end{aligned}
$$

For the Hessian model, $\mathrm{Hess}[\mathbf{H}^{(0)}] \in \mathbb{R}^{|\mathcal{V}| \times f \times d \times d}$,

$$\left(\mathrm{Hess}[\mathbf{H}^{(0)}]\right)_{i;g;kl} = \left(\frac{\partial}{\partial x_k} \frac{\partial}{\partial x_l} \mathbf{H}^{(0)}\right)_{i;g;} \tag{39}$$

$$= \sum_{j,m} D_{ij;;k}[D_{jm;;l}(H_{m;g;}^{(0)} - H_{l;g;}^{(0)}) - D_{im;;l}(H_{m;g;}^{(0)} - H_{i;g;}^{(0)})] \tag{40}$$

$$= \left[(\tilde{\mathbf{D}} \otimes \tilde{\mathbf{D}}) * \mathbf{H}^{(0)}\right]_{i;g;kl}. \tag{41}$$

## D IsoGCN MODELING DETAILS

To achieve isometric transformation invariance and equivariance, there are several rules to follow. Here, we describe the desired focus when constructing an IsoGCN model. In this section, a rank-$p$ tensor denotes a tensor the rank of which is $p \geq 1$ and $\sigma$ denotes a nonlinear activation function. $\boldsymbol{W}$ is a trainable weight matrix and $\boldsymbol{b}$ is a trainable bias.

### D.1 ACTIVATION AND BIAS

As the nonlinear activation function is not isometric transformation equivariant, nonlinear activation to rank-$p$ tensors cannot be applied, while one can apply any activation to rank-0 tensors. In addition, adding bias is also not isometric transformation equivariant, one cannot add bias when performing an affine transformation to rank-$p$ tensors. Again, one can add bias to rank-0 tensors.

Thus, for instance, if one converts from rank-0 tensors $\mathbf{H}^{(0)}$ to rank-1 tensors using IsoAM $\mathbf{G}$, $\mathbf{G}*\sigma(\mathbf{H}^{(0)}\boldsymbol{W}+\boldsymbol{b})$ and $(\mathbf{G}*\sigma(\mathbf{H}^{(0)}))\boldsymbol{W}$ are isometric equivariant functions, however $(\mathbf{G}*\mathbf{H}^{(0)})\boldsymbol{W}+\boldsymbol{b}$ and $\sigma\left((\mathbf{G} * \sigma(\mathbf{H}^{(0)}))\boldsymbol{W}\right)$ are not due to the bias and the nonlinear activation, respectively. Likewise, regarding a conversion from rank-1 tensors $\mathbf{H}^{(1)}$ to rank-0 tensors, $\sigma\left((\mathbf{G} \odot \mathbf{H}^{(1)})\boldsymbol{W} + \boldsymbol{b}\right)$ and $\sigma\left(\mathbf{G} \odot (\mathbf{H}^{(1)}\boldsymbol{W})\right)$ are isometric transformation invariant functions; however, $\mathbf{G} \odot (\mathbf{H}^{(1)}\boldsymbol{W} + \boldsymbol{b})$ and $(\mathbf{G} \odot \sigma(\mathbf{H}^{(1)}))\boldsymbol{W} + \boldsymbol{b}$ are not.

To convert rank-$p$ tensors to rank-$q$ tensors ($q \geq 1$), one can apply neither bias nor nonlinear activation. To add nonlinearity to such a conversion, we can multiply the converted rank-0 tensors $\sigma((\bigotimes^p \mathbf{G} \odot \mathbf{H}^{(p)})\boldsymbol{W} + \boldsymbol{b})$ with the input tensors $\mathbf{H}^{(p)}$ or the output tensors $\mathbf{H}^{(q)}$.

### D.2 PREPROCESSING OF INPUT FEATURE

Similarly to the discussion regarding the biases, we have to take care of the preprocessing of rank-$p$ tensors to retain isometric transformation invariance because adding a constant array and component-wise scaling could distort the tensors, resulting in broken isometric transformation equivariance.

For instance, $\mathbf{H}^{(p)}/\mathrm{Std}_{\mathrm{all}}\left[\mathbf{H}^{(p)}\right]$ is a valid transformation to retain isometric transformation equivariance, assuming $\mathrm{Std}_{\mathrm{all}}\left[\mathbf{H}^{(p)}\right] \in \mathbb{R}$ is a standard deviation of all components of $\mathbf{H}^{(p)}$. However, conversions such as $\mathbf{H}^{(p)}/\mathrm{Std}_{\mathrm{component}}\left[\mathbf{H}^{(p)}\right]$ and $\mathbf{H}^{(p)} - \mathrm{Mean}\left[\mathbf{H}^{(p)}\right]$ are not isometric transformation equivariant, assuming that $\mathrm{Std}_{\mathrm{component}}\left[\mathbf{H}^{(p)}\right] \in \mathbb{R}^{d^p}$ is a component-wise standard deviation.

### D.3 SCALING

Because the concrete instance of IsoAM $\tilde{\mathbf{D}}$ corresponds to the differential operator, the scale of the output after operations regarding $\tilde{D}$ can be huge. Thus, we rescale $\tilde{\mathbf{D}}$ using the scaling factor

$\left[\text{Mean}_{\text{sample},i}(\tilde{D}^2_{ii;;1} + \tilde{D}^2_{ii;;2} + \tilde{D}^2_{ii;;3})\right]^{1/2}$, where $\text{Mean}_{\text{sample},i}$ denotes the mean over the samples and vertices.

### D.4 IMPLEMENTATION

Because an adjacency matrix $\boldsymbol{A}$ is usually a sparse matrix for a regular mesh, $\boldsymbol{A}(m)$ in equation 16 is also a sparse matrix for a sufficiently small $m$. Thus, we can leverage sparse matrix multiplication in the IsoGCN computation. This is one major reason why IsoGCNs can compute rapidly. If the multiplication (tensor product or contraction) of IsoAMs must be computed multiple times the associative property of the IsoAM can be utilized.

For instance, it is apparent that $\left[\bigotimes^k \mathbf{G}\right] * \mathbf{H}^{(0)} = \mathbf{G} \otimes (\mathbf{G} \otimes \dots (\mathbf{G} * \mathbf{H}^{(0)}))$. Assuming that the number of nonzero elements in $\boldsymbol{A}(m)$ equals $n$ and $\mathbf{H}^{(0)} \in \mathbb{R}^{|\mathcal{V}| \times f}$, then the computational complexity of the right-hand side is $\mathcal{O}(n|\mathcal{V}|fd^k)$. This is an exponential order regarding $d$. However, $d$ and $k$ are usually small numbers (typically $d = 3$ and $k \leq 4$). Therefore one can compute an IsoGCN layer with a realistic spatial dimension $d$ and tensor rank $k$ fast and memory efficiently. In our implementation, both a sparse matrix operation and associative property are utilized to realize fast computation.

## E EXPERIMENT DETAILS: DIFFERENTIAL OPERATOR DATASET

### E.1 MODEL ARCHITECTURES

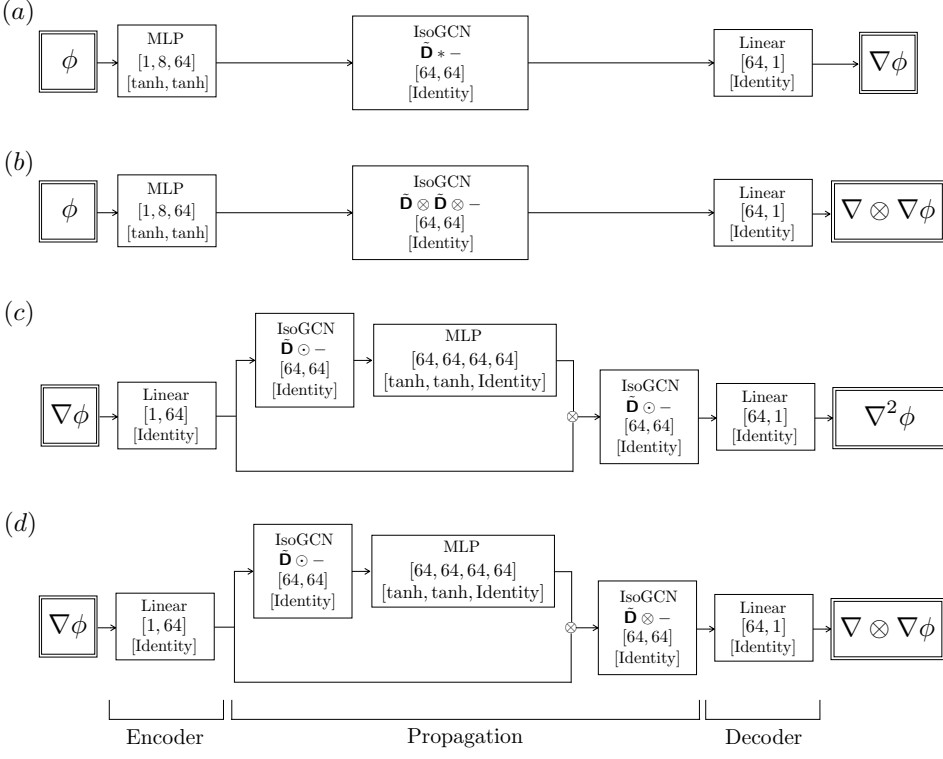

Figure 4: The IsoGCN model used for (a) the scalar field to the gradient field, (b) the scalar field to the Hessian field, (c) the gradient field to the Laplacian field, (d) the gradient field to the Hessian field of the gradient operator dataset. The numbers in each box denote the number of units. Below the unit numbers, the activation function used for each layer is also shown. $\otimes$ denotes the multiplication in the feature direction.

Table 5: Summary of the hyperparameter setting for both the TFN and SE(3)-Transformer. For the parameters not in the table, we used the default setting in the implementation of `https://github.com/FabianFuchsML/se3-transformer-public`.

|  | $0 \rightarrow 1$ | $0 \rightarrow 2$ | $1 \rightarrow 0$ | $1 \rightarrow 2$ |
|---|---|---|---|---|
| # hidden layers | 1 | 1 | 1 | 1 |
| # NL layers in the self-interaction | 1 | 1 | 1 | 1 |
| # channels | 24 | 20 | 24 | 24 |
| # maximum rank of the hidden layers | 1 | 2 | 1 | 2 |
| # nodes in the radial function | 16 | 8 | 16 | 22 |

Figure 4 represents the IsoGCN model used for the differential operator dataset. We used the `tanh` activation function as a nonlinear activation function because we expect the target temperature field to be smooth. Therefore, we avoid using non-differentiable activation functions such as the rectified linear unit (ReLU) (Nair & Hinton, 2010). For GCN and its variants, we simply replaced the IsoGCN layers with the corresponding ones. We stacked $m$ $(= 2, 5)$ layers for GCN, GIN, GCNII, and Cluster-GCN. We used an $m$ hop adjacency matrix for SGCN.

For the TFN and SE(3)-Transformer, we set the hyperparameters to have almost the same number of parameters as in the IsoGCN model. The settings of the hyperparameters are shown in Table 5.

### E.2 RESULT DETAILS

Table 6 represents the detailed comparison of training results. The results show that an IsoGCN outperforms other GCN models for all settings. Compared to other equivariant models, IsoGCN has competitive performance compared to equivariant models with shorter inference time as shown in Table 7. Therefore, it can be found out the proposed model has a strong expressive power to express differential regarding space with less computation resources compared to the TFN and SE(3)-Transformer.

## F EXPERIMENTS DETAILS: ANISOTROPIC NONLINEAR HEAT EQUATION DATASET

### F.1 DATASET

The purpose of the experiment was to solve the anisotropic nonlinear heat diffusion under an adiabatic boundary condition. The governing equation is defined as follows:

$$\Omega \subset \mathbb{R}^3, \tag{42}$$

$$\frac{\partial T(\boldsymbol{x}, t)}{\partial t} = \nabla \cdot \boldsymbol{C}(T(\boldsymbol{x}, t)) \nabla T(\boldsymbol{x}, t), \text{in } \Omega, \tag{43}$$

$$T(\boldsymbol{x}, t = 0) = T_{0.0}(\boldsymbol{x}), \text{in } \Omega, \tag{44}$$

$$\nabla T(\boldsymbol{x}, t)|_{\boldsymbol{x} = \boldsymbol{x}_b} \cdot \boldsymbol{n}(\boldsymbol{x}_b) = 0, \text{on } \partial \Omega, \tag{45}$$

where $T$ is the temperature field, $T_{0.0}$ is the initial temperature field, $\boldsymbol{C} \in \mathbb{R}^{d \times d}$ is an anisotropic diffusion tensor and $\boldsymbol{n}(\boldsymbol{x}_b)$ is the normal vector at $\boldsymbol{x}_b \in \partial \Omega$. Here, $\boldsymbol{C}$ depends on temperature thus the equation is nonlinear. We randomly generate $\boldsymbol{C}(T = -1)$ for it to be a positive semidefinite symmetric tensor with eigenvalues varying from 0.0 to 0.02. Then, we defined the linear temperature dependency the slope of which is $-\boldsymbol{C}(T = -1)/4$. The function of the anisotropic diffusion tensor is uniform for each sample. The task is defined to predict the temperature field at $t = 0.2, 0.4, 0.6, 1.0$ $(T_{0.2}, T_{0.4}, T_{0.6}, T_{0.8}, T_{1.0})$ from the given initial temperature field, material property, and mesh geometry. However, the performance is evaluated only with $T_{1.0}$ to focus on the predictive performance. We inserted other output features to stabilize the trainings. Accordingly, the diffusion number of this problem is $\boldsymbol{C} \Delta t / (\Delta x)^2 \simeq 10.0^4$ assuming $\Delta x \simeq 10.0^{-3}$.

Figure 5 represents the process of generating the dataset. We generated up to 9 FEA results for each CAD shape. To avoid data leakage in terms of the CAD shapes, we first split them into training, validation, and test datasets, and then applied the following process.

Table 6: Summary of the test losses (mean squared error $\pm$ the standard error of the mean in the original scale) of the differential operator dataset: $0 \rightarrow 1$ (the scalar field to the gradient field), $0 \rightarrow 2$ (the scalar field to the Hessian field), $1 \rightarrow 0$ (the gradient field to the Laplacian field), and $1 \rightarrow 2$ (the gradient field to the Hessian field). Here, if "$\boldsymbol{x}$" is "Yes", $\boldsymbol{x}$ is also in the input feature.

| Method | # hops | $\boldsymbol{x}$ | Loss of $0 \rightarrow 1$ $\times 10^{-5}$ | Loss of $0 \rightarrow 2$ $\times 10^{-6}$ | Loss of $1 \rightarrow 0$ $\times 10^{-6}$ | Loss of $1 \rightarrow 2$ $\times 10^{-6}$ |
|---|---|---|---|---|---|---|
| GIN | 2 | No | $151.19 \pm 0.53$ | $49.10 \pm 0.36$ | $542.52 \pm 2.14$ | $59.65 \pm 0.46$ |
|  | 2 | Yes | $147.10 \pm 0.51$ | $47.56 \pm 0.35$ | $463.79 \pm 2.08$ | $50.73 \pm 0.40$ |
|  | 5 | No | $151.18 \pm 0.53$ | $48.99 \pm 0.36$ | $542.54 \pm 2.14$ | $59.64 \pm 0.46$ |
|  | 5 | Yes | $147.07 \pm 0.51$ | $47.35 \pm 0.35$ | $404.92 \pm 1.74$ | $46.18 \pm 0.39$ |
| GCNII | 2 | No | $151.18 \pm 0.53$ | $43.08 \pm 0.31$ | $542.74 \pm 2.14$ | $59.65 \pm 0.46$ |
|  | 2 | Yes | $151.14 \pm 0.53$ | $40.72 \pm 0.29$ | $194.65 \pm 1.00$ | $45.43 \pm 0.36$ |
|  | 5 | No | $151.11 \pm 0.53$ | $32.85 \pm 0.23$ | $542.65 \pm 2.14$ | $59.66 \pm 0.46$ |
|  | 5 | Yes | $151.13 \pm 0.53$ | $31.87 \pm 0.22$ | $280.61 \pm 1.30$ | $39.38 \pm 0.34$ |
| SGCN | 2 | No | $151.17 \pm 0.53$ | $50.26 \pm 0.38$ | $542.90 \pm 2.14$ | $59.65 \pm 0.46$ |
|  | 2 | Yes | $151.12 \pm 0.53$ | $49.96 \pm 0.37$ | $353.29 \pm 1.49$ | $59.61 \pm 0.46$ |
|  | 5 | No | $151.12 \pm 0.53$ | $55.02 \pm 0.42$ | $542.73 \pm 2.14$ | $59.64 \pm 0.46$ |
|  | 5 | Yes | $151.16 \pm 0.53$ | $55.08 \pm 0.42$ | $127.21 \pm 0.63$ | $56.97 \pm 0.44$ |
| GCN | 2 | No | $151.23 \pm 0.53$ | $49.59 \pm 0.37$ | $542.54 \pm 2.14$ | $59.64 \pm 0.46$ |
|  | 2 | Yes | $151.14 \pm 0.53$ | $47.91 \pm 0.35$ | $542.68 \pm 2.14$ | $59.60 \pm 0.46$ |
|  | 5 | No | $151.18 \pm 0.53$ | $50.58 \pm 0.38$ | $542.53 \pm 2.14$ | $59.64 \pm 0.46$ |
|  | 5 | Yes | $151.14 \pm 0.53$ | $48.50 \pm 0.35$ | $542.30 \pm 2.14$ | $25.37 \pm 0.28$ |
| Cluster-GCN | 2 | No | $151.19 \pm 0.53$ | $33.39 \pm 0.24$ | $542.54 \pm 2.14$ | $59.66 \pm 0.46$ |
|  | 2 | Yes | $147.23 \pm 0.51$ | $32.29 \pm 0.24$ | $167.73 \pm 0.83$ | $17.72 \pm 0.17$ |
|  | 5 | No | $151.15 \pm 0.53$ | $28.79 \pm 0.21$ | $542.51 \pm 2.14$ | $59.66 \pm 0.46$ |
|  | 5 | Yes | $146.91 \pm 0.51$ | $26.60 \pm 0.19$ | $185.21 \pm 0.99$ | $18.18 \pm 0.20$ |
| TFN | 2 | No | $2.47 \pm 0.02$ | OOM | $26.69 \pm 0.24$ | OOM |
|  | 5 | No | OOM | OOM | OOM | OOM |
| SE(3)-Trans. | 2 | No | $\mathbf{1.79} \pm 0.02$ | $\mathbf{3.50} \pm 0.04$ | $\mathbf{2.52} \pm 0.02$ | OOM |
|  | 5 | No | $2.12 \pm 0.02$ | OOM | $7.66 \pm 0.05$ | OOM |
| **IsoGCN** (Ours) | 2 | No | $2.67 \pm 0.02$ | $6.37 \pm 0.07$ | $7.18 \pm 0.06$ | $\mathbf{1.44} \pm 0.02$ |
|  | 5 | No | $14.19 \pm 0.10$ | $21.72 \pm 0.25$ | $34.09 \pm 0.19$ | $8.32 \pm 0.09$ |

Table 7: Summary of the inference time on the test dataset. $0 \rightarrow 1$ corresponds to the scalar field to the gradient field, and $0 \rightarrow 2$ corresponds to the scalar field to the Hessian field. Each computation was run on the same GPU (NVIDIA Tesla V100 with 32 GiB memory). OOM denotes the out-of-memory of the GPU.

| Method | $0 \rightarrow 1$ | | $0 \rightarrow 2$ | |
|---|---|---|---|---|
|  | # parameters | Inference time [s] | # parameters | Inference time [s] |
| TFN | 5264 | 3.8 | 5220 | OOM |
| SE(3)-Trans. | 5392 | 4.0 | 5265 | 9.2 |
| **IsoGCN** (Ours) | 4816 | 0.4 | 4816 | 0.7 |

Using one CAD shape, we generated up to three meshes using clscale (a control parameter of the mesh characteristic lengths) = 0.20, 0.25, and 0.30. To facilitate the training process, we scaled the meshes to fit into a cube with an edge length equal to 1.

Using one mesh, we generated three initial conditions randomly using a Fourier series of the 2nd to 10th orders. We then applied an FEA to each initial condition and material property determined randomly as described above. We applied an implicit method to solve time evolutions and a direct method to solve the linear equations. The FEA time step $\Delta t$ was set to 0.01.

During this process, some of the meshes or FEA results may not have been available due to excessive computation time or non-convergence. Therefore, the size of the dataset was not exactly equal to the number multiplied by 9. Finally, we obtained 439 FEA results for the training dataset, 143 FEA results for the validation dataset, and 140 FEA results for the test dataset.

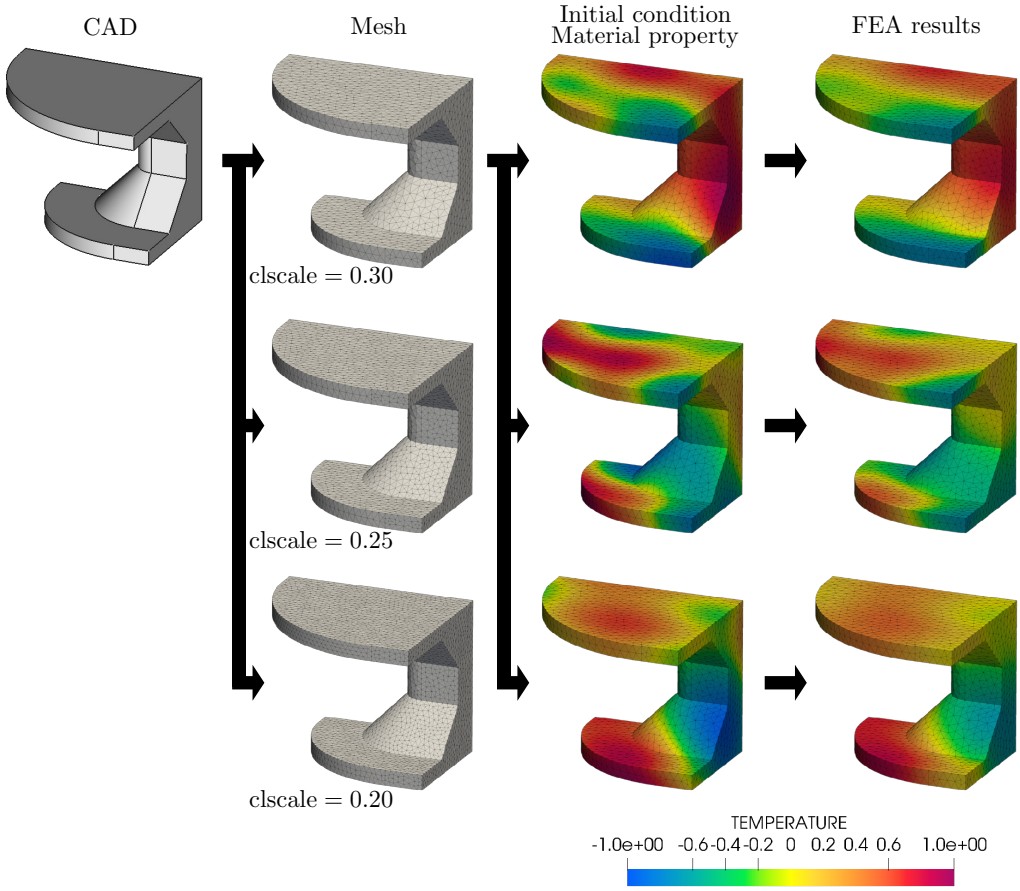

Figure 5: The process of generating the dataset. A smaller clscale parameter generates smaller meshes.

### F.2 INPUT FEATURES

To express the geometry information, we extracted the effective volume of the $i$th vertex $V_i^{\text{effective}}$ and the mean volume of the $i$th vertex $V_i^{\text{mean}}$, which are defined as follows:

$$V_i^{\text{effective}} = \sum_{e \in \mathcal{N}_i^e} \frac{1}{4} V_e, \tag{46}$$

$$V_i^{\text{mean}} = \frac{\sum_{e \in \mathcal{N}_i^e} V_e}{|\mathcal{N}_i^e|}, \tag{47}$$

where $\mathcal{N}_i^e$ is the set of elements, including the $i$th vertex.

For GCN or its variant models, we tested several combinations of input vertex features $T_{0.0}$, $\boldsymbol{C}$, $V^{\text{effective}}$, $V^{\text{mean}}$, and $\boldsymbol{x}$ (Table 9). For the IsoGCN model, inputs were $T_{0.0}$, $\boldsymbol{C}$, $V^{\text{effective}}$, and $V^{\text{mean}}$.

### F.3 MODEL ARCHITECTURES

Figure 6 represents the IsoGCN model used for the anisotropic nonlinear heat equation dataset. We used the $\tanh$ activation function as a nonlinear activation function because we expect the target temperature field to be smooth. Therefore, we avoid using non-differentiable activation functions such as the rectified linear unit (ReLU) (Nair & Hinton, 2010). Although the model looks complicated, one propagation block corresponds to the first-order Taylor expansion $T(t+\Delta t) \simeq \nabla \boldsymbol{C} \odot \nabla T(t) + T(t)$ because the propagation block is expressed as $\tilde{\boldsymbol{D}} \odot \boldsymbol{C} \odot \text{MLP}(T)\tilde{\boldsymbol{D}} * T + T$, where $T$ denotes the

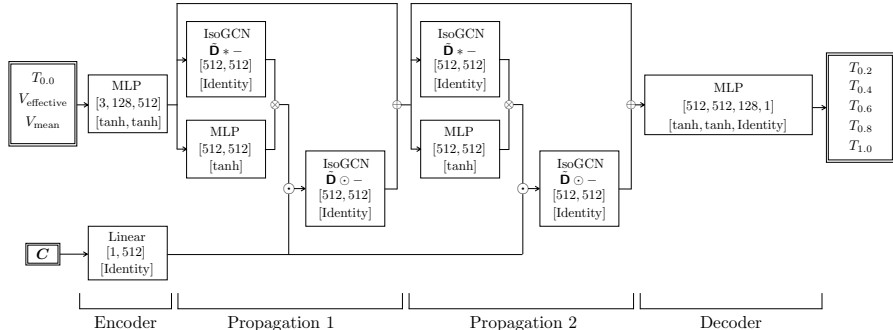

Figure 6: The IsoGCN model used for the anisotropic nonlinear heat equation dataset. The numbers in each box denote the number of units. Below the unit numbers, the activation function used for each layer is also shown. $\otimes$ denotes multiplication in the feature direction, $\odot$ denotes the contraction, and $\oplus$ denotes the addition in the feature direction.

Table 8: Summary of the hyperparameter setting for both the TFN and SE(3)-Transformer. For the parameters not written in the table, we used the default setting in the implementation of `https://github.com/FabianFuchsML/se3-transformer-public`.

| | |
|---|---|
| # hidden layers | 1 |
| # NL layers in the self-interaction | 1 |
| # channels | 16 |
| # maximum rank of the hidden layers | 2 |
| # nodes in the radial function | 32 |

rank-0 tensor input to the propagation block. By stacking this propagation block $p$ times, we can approximate the $p$th order Taylor expansion of the anisotropic nonlinear heat equation.

For GCN and its variants, we simply replaced the IsoGCN layers with the corresponding ones. We stacked $m \, (= 2, 5)$ layers for GCN, GIN, GCNII, and Cluster-GCN. We used an $m$ hop adjacency matrix for SGCN.

For the TFN and SE(3)-Transformer, we set the hyperparameters to as many parameters as possible that would fit on the GPU because the TFN and SE(3)-Transformer with almost the same number of parameters as in IsoGCN did not fit on the GPU we used (NVIDIA Tesla V100 with 32 GiB memory). The settings of the hyperparameters are shown in Table 8.

## F.4 RESULT DETAILS

Table 9 shows a detailed comparison of the training results. The inclusion of $x$ in the input features of the baseline models did not improve the performance. In addition, if $x$ is included in the input features, a loss of the generalization capacity for larger shapes compared to the training dataset may result as it extrapolates. The proposed model achieved the best performance compared to the baseline models considered. Therefore, we concluded that the essential features regarding the mesh shapes are included in $\tilde{\mathbf{D}}$. Besides, IsoGCN can scale up to meshes with 1M vertices as shown in Figure 7.

Table 9: Summary of the test losses (mean squared error $\pm$ the standard error of the mean in the original scale) of the anisotropic nonlinear heat dataset. Here, if "$x$" is "Yes", $x$ is also in the input feature. OOM denotes the out-of-memory on the applied GPU (32 GiB).

| Method | # hops | $x$ | Loss $\times 10^{-3}$ |
|---|---|---|---|
| GIN | 2 | No | $16.921 \pm 0.040$ |
| | 2 | Yes | $18.483 \pm 0.025$ |
| | 5 | No | $22.961 \pm 0.056$ |
| | 5 | Yes | $17.637 \pm 0.046$ |
| GCN | 2 | No | $10.427 \pm 0.028$ |
| | 2 | Yes | $11.610 \pm 0.032$ |
| | 5 | No | $12.139 \pm 0.031$ |
| | 5 | Yes | $11.404 \pm 0.032$ |
| GCNII | 2 | No | $9.595 \pm 0.026$ |
| | 2 | Yes | $9.789 \pm 0.028$ |
| | 5 | No | $8.377 \pm 0.024$ |
| | 5 | Yes | $9.172 \pm 0.028$ |
| Cluster-GCN | 2 | No | $7.266 \pm 0.021$ |
| | 2 | Yes | $8.532 \pm 0.023$ |
| | 5 | No | $8.680 \pm 0.024$ |
| | 5 | Yes | $10.712 \pm 0.030$ |
| SGCN | 2 | No | $7.317 \pm 0.021$ |
| | 2 | Yes | $9.083 \pm 0.026$ |
| | 5 | No | $6.426 \pm 0.018$ |
| | 5 | Yes | $6.519 \pm 0.020$ |
| TFN | 2 | No | $15.661 \pm 0.019$ |
| | 5 | No | OOM |
| SE(3)-Trans. | 2 | No | $14.164 \pm 0.018$ |
| | 5 | No | OOM |
| **IsoGCN** (Ours) | 2 | No | $4.674 \pm 0.014$ |
| | 5 | No | $\mathbf{2.470} \pm 0.008$ |

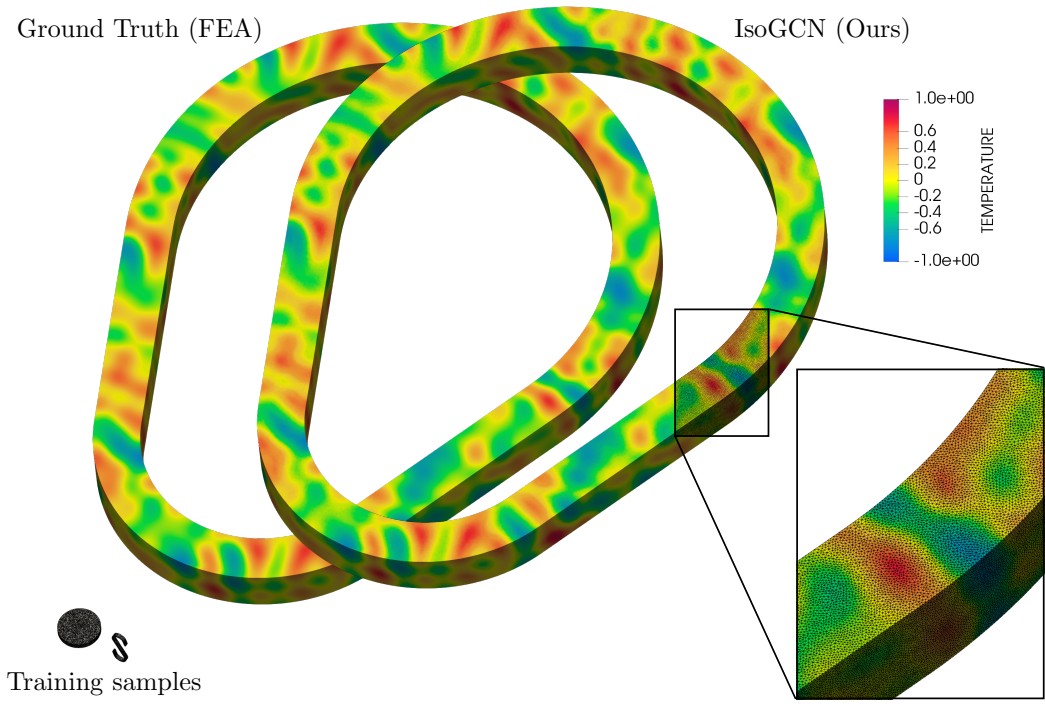

Figure 7: Comparison between (left) samples in the training dataset, (center) ground truth computed through FEA, and (right) IsoGCN inference result. For both the ground truth and inference result, $|\mathcal{V}| = 1,011,301$. One can see that IsoGCN can predict the temperature field for a mesh, which is much larger than these in the training dataset.

