# OpenReview forum: "Isometric Transformation Invariant and Equivariant Graph Convolutional Networks"
_ICLR.cc/2021/Conference — ICLR 2021 Poster_

### Official Review · AnonReviewer3 · 2020-10-25
**Isometric Transformation Invariant and Equivariant Graph Convolutional Networks. Summary: This paper proposed a transformation invariant and equivariant GCN, which mainly focused on the construction of the newly defined isometric adjacency matrix. Apply this isometric adjacency matrix to change the formulation of the GCN proposed by Kipf & Welling.**

**Rating:** 5
**Confidence:** 3

**Review:**

Comments:
Pros:
1.	This paper gives a comprehensive derivation of propositions used in the construction of isoGCN.
2.	The properties provided are clear
3.	The idea to realize the purpose of being isometric transformation invariant and equivariant is good.

Cons:
1.	More explanations about the superiority of isometric transformation invariance and equivariance need to be addressed since this purpose seems quite important throughout this paper. Besides, how this purpose helps to achieve the superiority needs to be answered.
2.	How to interpret the adjacency matrix up to m hops, what is the construction of the adjacency matrix mentioned in the paper?
3.	In the experiment results, though the proposed isoGCN gave the minimum loss when comparing with other GCN variants, the scale of loss is around 10^(-5/-6) which may not give a significant goodness.
4.	In table 4, though isoGCN gave the least running time, it also gave larger loss compared with the method FrontISTR. How to explain the trade-off between running time and the loss?
5.     The paper is too theoretical and does not fit to ICLR.  Real impact and real application should be emphasized more.

---

> ### Author Response · Authors · 2020-11-24
> **Response to AnonReviewer3**
>
> We thank the reviewer for the time and effort in reviewing our work. We respond to the points in the review.
>
> Q. More explanations about the superiority of isometric transformation invariance and equivariance need to be addressed, since this purpose seems quite important throughout this paper. Besides, how this purpose helps to achieve the superiority needs to be answered.
>
> A. We added a discussion regarding the superiority of isometric transformation invariance and equivariance (sec 1). In the literature, many benefits have been discussed, such as high interpretability due to the shared weight, stability, and predictability of the model output, and the efficiency of training due to data augmentation not being required. All these points are essential, especially for learning physical simulations because every physical quantity and physical law is either isometric transformation invariant or equivariant.
>
> In real applications, physical simulations can be performed under various isometric transformations. For instance, if we imagine simulating every window on a big building, each window has the same shape but a different position and thus is transformed isometrically. If we use a model without equivariance, when facing different inference results for windows A and B, we cannot determine why they are different because it may be due to the physical condition or lack of model equivariance. Using the equivariant model, one can determine that the physical condition is different and makes more reliable decisions. Thus, the model's stability and predictability are particularly important in the real application to physical simulations.
>
> Q. How to interpret the adjacency matrix up to m hops, what is the construction of the adjacency matrix mentioned in the paper?
>
> A. We added the definition of the adjacency matrix up to m hops and adjacency. Because we deal with mesh-structured data, we define two nodes that are adjacent when they share the same cell, which is commonly assumed in physical simulation methods (e.g., finite volume method and finite element method).
>
> Q. In the experiment results, though the proposed isoGCN gave the minimum loss when comparing with other GCN variants, the scale of loss is around 10^(-5/-6), which may not give a significant goodness.
>
> A. This is because we evaluated loss in the original scale (although we performed training with scaling). We prefer to evaluate at the original scale because it can be more intuitive, especially when dealing with physical data. Nevertheless, the absolute value has less meaning when comparing model performances. Looking at the standard error of the mean and qualitative comparison, we clearly see that IsoGCN is significantly better than other GCN-like models.
>
> Q. In table 4, though isoGCN gave the least running time, it also gave larger loss compared with the method FrontISTR. How to explain the trade-off between running time and the loss?
>
> A. In fact, the IsoGCN loss is smaller than that of the FrontISTR with $\Delta t = 1.0$ for $|V| = 21,289$ and $155,019$ cases. Compared to FrontISTR with $\Delta t = 0.5$, IsoGCN has a larger loss, as the reviewer pointed out. Because FrontISTR calculates the inverse matrix obtained from the partial differential equation, it can calculate interactions between vertices that are distant from each other. In contrast, IsoGCN looks only in a limited area to compute the next state of the considered vertex. Of course, it is possible to increase the number of hops considered by the IsoGCN, but this will lead to extending the computation time. To answer the question, the trade-off between running time and the loss is found here. Considering only a part of the graph (like IsoGCN) tends to lead to shorter computation time and moderate accuracy. Conversely, considering a huge or entire part of the graph (such as FrontISTR) tends to lead to longer computation time and high accuracy (this is also because FrontISTR is the ground truth).
>
> Q. The paper is too theoretical and does not fit to ICLR. Real impact and real application should be emphasized more.
>
> A. As pointed out, our work has a strong background in theory. However, IsoGCN demonstrates a real impact, that is, high accuracy and high stability due to the isometric transformation invariance and equivariance in addition to the significantly shorter computation time compared to a physical simulation, which was impossible with the existing equivariant models. Suppose the computation time of a machine learning model is longer than that of a physical simulation. In this case, the benefit of the machine learning model degrades because one has no reason to prefer the machine learning model to the physical simulation. Therefore, we can say that IsoGCN is the first model to learn physical simulations "properly," in terms of both equivariance (please see the answer to the first question) and fast computation time.

---

### Official Review · AnonReviewer5 · 2020-11-04
**Strong core idea, but unclear section 3 and missing related work and baselines**

**Rating:** 7
**Confidence:** 4

**Review:**

Summary:
The paper proposes a network that operates on features of graphs that are embedded in a d-dim Euclidean space. The paper considers equivariance to a group G that is the direct product of permutations of N points and Euclidean transformations. The features they consider are tensor products of the N-dimensional natural representations of permutations and the d-dimensional standard representation of O(d). From the coordinates, an “isometric adjacency matrix” is created, which is such a tensor. This matrix is combined in various G-equivariant ways with the features and then linearly combined with learnable weights to create new features. These operations are interleaved with non-linearities to form the network. The authors compare to several graph network methods and show competitive performance on several tasks.

Strengths:
-	The authors build a bridge between the global permutation equivariant methods on graphs, as they consider matrix-like features and equivariant maps between them, and Euclidean-equivariant methods on point clouds. This combination seems a powerful approach to point-cloud neural networks.

Weaknesses:
-	Important pieces of prior work are missing from the related work section. The paper seems to be strongly related to Tensor Field Networks (TFN) (Thomas et al. 2018), as both define Euclidean and permutation equivariant convolutions on point clouds / graphs. Furthermore, there are several other methods that operate on graph that are embedded in a Euclidean space, such as SchNet (Schütt et al 2017). The graph network methods currently discussed all do not include the point coordinates in their operations. Lastly, the proposed method operates globally linearly on features on a graph, equivariantly to permutations, which is done in prior work, e.g. Maron 2018.
-	The experimental section only compares to methods that in their convolution are unaware of the point coordinates (except for in the input features). A comparison to coordinate-aware methods, such as TFN or SchNet seems appropriate.
-	The core object, the isometric adjacency matrix G, is ill-defined. In Eq 1 it is defined trough the embedding coordinates and “the transformation invariant rank-2 tensor” T. This object is not defined in the paper, which makes section 3 very confusing to read. In section 3, it appears like that the defined objects D take the role of object G in the above, so what is the role of eq 1?
-	In section 3, the authors speak of “collections of rank-p tensors”. However, these objects seem to actually be tensors of the shape N^a x d^p, where N is the number of nodes, d is the dimensionality of the embedding, and a and p are natural numbers. These objects transform under both permutations and Euclidean transformations in the obvious way. Why not make this fact explicit? That would make section 3 much easier to read. It seems like that when p=0, then a=1, and when p>0, then a=2.  Except for in sec 3.2.2, in which a p=3 tensor has a=1.
-	In Sec 3.2, what are f_in and f_out? Are these the dimensionalities of the tensor product representation? Or do they denote the number of copies of the representation? If it’s the former, I don’t see how the network is equivariant. If it’s the latter, I don’t understand the last paragraph of 3.2.2, which says 1H \in R^{N x f_in}, which looks like a 0-tensor.
-	Can the authors clarify “To achieve translation equivariance, a constant tensor can be added to the output collection of tensors.”? The proposed method seems to only lead to translation invariant features. I do not follow how adding a constant tensor leads to translation equivariance that is not invariance.
-	Am I correct in understanding that the method scales cubic with the number of vertices (e.g. eqs 4, 6)? Or is there some sparsity used in the implementation, but not mentioned? Should we expect a method of cubic complexity to scale to 1M vertices? In a naïve implementation, a fast modern GPU with 14.2E12 flops would need 20h for a single 1Mx1M matrix-matrix multiplication (1E18 floating point operations).
-	The authors claim the method scales to 1M vertices, but I cannot find this in the experiments. Table 4 speaks of 155k vertices. How did the authors determine the method scales to 1M vertices?

Recommendation:
In its current form, I recommend rejection of this paper. Section 3 is insufficiently clear written, the related work lack important references to prior work and the experiments lack a comparison to potentially strong other methods. This is a shame, because I’d like to see this paper succeed, as the core idea is very strong. Significant improvements in the above criticisms can improve my score.

Suggestions for improvement:
-	Be clear about what the G object is and what eq 1 means.
-	Be explicit about types the objects, be more explicit about the indices that refer to the permutation representation, to the indices that refer to the Euclidean representation and the indices that refer to copies of the same representation. I think there is an opportunity to be more clear, more explicit, while reducing notational clutter.
-	Expand the related work section
-	Compare to the strong baselines that use the coordinates.
-	Provide argumentation for the claim to scale to 1M vertices.

Minor points:
-	Eq 7, \times should be \otimes?
-	Eq 14, what is j?
-	The authors write: “A, B and C are X, Y and Z respectively”. Perhaps this could be re-written to the easier to read “A=X, B=Y and C=Z”. This happens each time the word “respectively” is used.
-	Table 3 typo, gluster -> cluster


### Post rebuttal
The authors addressed all my concerns and strongly improved their paper. I think it is now a good candidate for acceptance, as it provides an interesting alternative to / variation on tensor field networks. I raise my rating from 4 to 7.

---

> ### Author Response · Authors · 2020-11-24
> **Response to AnonReviewer5 (part 1)**
>
> We thank the reviewer for the time and effort in reviewing our work. We respond to the points in the review.
>
> Q. Important pieces of prior work are missing from the related work section. The paper seems to be strongly related to Tensor Field Networks (TFN) (Thomas et al. 2018), as both define Euclidean and permutation equivariant convolutions on point clouds / graphs. Furthermore, there are several other methods that operate on graph that are embedded in a Euclidean space, such as SchNet (Schütt et al 2017). The graph network methods currently discussed all do not include the point coordinates in their operations. Lastly, the proposed method operates globally linearly on features on a graph, equivariantly to permutations, which is done in prior work, e.g. Maron 2018.
>
> A. We added a reference to Maron et al. 2018 (Section 3.2) and removed the proof regarding permutation equivariance because this is already done, as you pointed out.
>
> Q. The experimental section only compares to methods that in their convolution are unaware of the point coordinates (except for in the input features). A comparison to coordinate-aware methods, such as TFN or SchNet seems appropriate.
>
> A. Thank you for the suggestion. We added tensor field networks (Thomas et al. 2018) and SE(3)-Transformers (Fuchs et al. 2020) to the baseline models in the experiments (Section 4), and IsoGCN showed competitive performance against them and significantly faster computation time, which supports our claim.
>
> Q. The core object, the isometric adjacency matrix G, is ill-defined. In Eq 1 it is defined trough the embedding coordinates and "the transformation invariant rank-2 tensor" T. This object is not defined in the paper, which makes section 3 very confusing to read. In section 3, it appears like that the defined objects D take the role of object G in the above, so what is the role of eq 1?
>
> A. We added explanations and figures in Section 3.1. T can be determined by users, as long as it meets the requirement (tensor rank and non-trainability). The point is that we would like to keep the assumption as small as possible to demonstrate that the concept of IsoGCN applies to a wide range of graphs. Thus, we have retained the definition of G abstract. In contrast, D is a specific form of G, but D can be obtained validly only for a graph derived from a mesh. Otherwise, we cannot define $M^{-1}$ in a general graph.
>
> Q. In section 3, the authors speak of "collections of rank-p tensors". However, these objects seem to actually be tensors of the shape N^a x d^p, where N is the number of nodes, d is the dimensionality of the embedding, and a and p are natural numbers. These objects transform under both permutations and Euclidean transformations in the obvious way. Why not make this fact explicit? That would make section 3 much easier to read. It seems like that when p=0, then a=1, and when p>0, then a=2. Except for in sec 3.2.2, in which a p=3 tensor has a=1.
>
> A. Thank you for the suggestion. We have updated our notation to use "tensor fields" instead of "collections of tensors." At the same time, we clarified the representation of each index, i.e., a rank-p tensor field is indexed $H_{i; g; k_1 k_2 ... k_p}$, where i corresponds to permutation, g denotes the feature index, and k_1, k_2, ..., k_p correspond to spatial indices. $G$ is indexed $G_{i, j;; k}$, where i and  j correspond to permutation, and k corresponds to spatial index. These are detailed at the beginning of Section 3 and 3.1.
>
> Q. In Sec 3.2, what are f_in and f_out? Are these the dimensionalities of the tensor product representation? Or do they denote the number of copies of the representation? If it's the former, I don't see how the network is equivariant. If it's the latter, I don't understand the last paragraph of 3.2.2, which says 1H \in R^{N x f_in}, which looks like a 0-tensor.
>
> A. There were typos in that section; we intended f_in and f_out to be the number of features and a rank-3 tensor field should be in the shape of $R^{|V| \times f_\mathrm{in} \times d^3}$ (in case of input). We corrected them and it now should be clear.

---

> > ### Author Response · Authors · 2020-11-24
> > **Response to AnonReviewer5 (part 2)**
> >
> > Q. Can the authors clarify "To achieve translation equivariance, a constant tensor can be added to the output collection of tensors."? The proposed method seems to only lead to translation invariant features. I do not follow how adding a constant tensor leads to translation equivariance that is not invariance.
> >
> > A. We added clarification of that point at the end of Section 3.2.2. The point is to define x_ref as a reference point and then added it to an entire graph, which makes the graph transformed.
> >
> > Q. Am I correct in understanding that the method scales cubic with the number of vertices (e.g. eqs 4, 6)? Or is there some sparsity used in the implementation, but not mentioned? Should we expect a method of cubic complexity to scale to 1M vertices? In a naïve implementation, a fast modern GPU with 14.2E12 flops would need 20h for a single 1Mx1M matrix-matrix multiplication (1E18 floating point operations).
> >
> > A. We typically use sparse matrices; thus the complexity does not scale cubically with the number of vertices. Appendix D.4 was updated to mention the sparsity of IsoAMs derived from meshes and complexity analysis. In our implementation (which is uploaded as supplementary material), the order scales cubically with the spatial dimension, which is usually small (e.g., 3).
> >
> > Q. The authors claim the method scales to 1M vertices, but I cannot find this in the experiments. Table 4 speaks of 155k vertices. How did the authors determine the method scales to 1M vertices?
> >
> > A. We updated Table 4 to show the results with 1M vertices. The reason for not having them in the first version was that the computation time of the ground truth was too long to meet the deadline.
> >
> > Q. Recommendation: In its current form, I recommend rejection of this paper. Section 3 is insufficiently clear written, the related work lack important references to prior work and the experiments lack a comparison to potentially strong other methods. This is a shame, because I'd like to see this paper succeed, as the core idea is very strong. Significant improvements in the above criticisms can improve my score.
> >
> > A. Thank you for your positive comment. We refined Section 3 and added equivariant baseline models, which made improved our paper. We hope that our update resolves all the issues. Also, thank you for your suggestions. We incorporated them as much as possible and now think that our paper is much the better for it.
> >
> > Q. Minor points
> >
> > A. All are fixed (j in Eq 14 (Eq 16 in the updated version) should be l), while we retained a few "respectively"s when it sounds natural. Thank you again for the encouraging feedback.

---

### Official Review · AnonReviewer4 · 2020-11-06
**Equivariant GCN with lacking clarity**

**Rating:** 6
**Confidence:** 3

**Review:**

The paper proposes a graph convolutional network that can be in-/equivariant to isometric transformation. The method is applied to a linear toy problem of predicting the result of several diiferential operators and a nonlinear heat diffusion dataset in comparison to finite element analysis (FEA).

A major problem of the paper is its lack of clarity, which is largely due to the unusual definition of established terms, for example:
- The contraction (Eq. 4) is usually an operator defined for a single tensor, not two.
- It is not completely clear to me how Eq. 3 is related to the convolution. Summing over index j look similar to a typical graph convolution, however, here you have an additional index l. Written loosely, a convolution usually has the form $\sum_j f(i, j) g(j) $, but here you have $\sum_j f(i, j) g(j, l)$.
- The notion of a "collection of tensors" does not work well here, in particular when aiming to define a convolution (over a space or group). Given the particular application, the notion of tensor fields might be more appropriate.

The authors mention previous work on equivariant neural networks such as steerable convolutions, tensor field networks, and covariant networks, which they claim are inefficient since they use message-passing, while their approach uses GCNs. This contradicts their statement from above, that GCNs are message-passing neural networks. It should be clarified what exactly makes these approaches less efficient. Moreover, since equivariant neural networks are clearly more suitable for the presented experiments then previous GCNs, some of them should be included in the model comparisons with timings. In particular, going to tensors with large rank (as suggested in Eqs. 8-10) should be rather inefficient compared to equivariant networks using spherical harmonics due to the exponentially increasing number of parameters.

Regarding the timings in Table 4, it is not clear to me, whether the timing include the preprocessing of the adjacency matrix for the inference. Also the caption mentions that a single CPU core was used. Does this refer to both the neural network and the FEA? Otherwise it does not seem like fair comparison.

Pros
====
- incorporating the various differential operators in the network structures in Sec 3.3 is an interesting idea

Cons
====
- lacking clarity in method description and notation
- conceptual differences to steerable convolutions and equivariant networks do not become sufficiently clear
- experiment section not very strong since comparison to equivariant networks is missing and the timings in Table 4 need some clarification

Update: I read the comments of the authors and thank them for the clarifications. The additional baselines improved the paper. I raised my score to reflect that.

---

> ### Author Response · Authors · 2020-11-24
> **Response to AnonReviewer4**
>
> We thank the reviewer for the time and effort in reviewing our work. We respond to the points raised in the review.
>
> Q. The contraction (Eq. 4) is usually an operator defined for a single tensor, not two.
>
> A. As pointed out, the contraction can be defined for a single tensor, but at the same time, the contraction between two or more tensors can be defined. For instance, please see p.4 in [1] or Eq. (4.22), (5.7) in [2]. Therefore, we concluded that our usage of "contraction" is common.
>
>
> Q. It is not completely clear to me how Eq. 3 is related to the convolution. Summing over index j look similar to a typical graph convolution, however, here you have an additional index l. Written loosely, a convolution usually has the form $\sum_j f(i,j)g(j)$ , but here you have $\sum_j f(i,j) g(j,l)$ .
>
> A. Our definition of convolution is based on GCN (Kipf et al. 2017). In GCN, we have $\sum_j A_{ij} H_{jl}$, which is quite analogous to our definition. In fact, denoting $g(i, 1) = x(i), g(i, 2) = y(i), \dots,$ we can confirm that $\sum_j f(i,j) g(j,l)$ is a generalization of $\sum_j f(i,j)g(j)$ . In this paper, we emphasize the relationship between our convolution and that in the GCN (Section 3.1).
>
> Q. The notion of a "collection of tensors" does not work well here, in particular when aiming to define a convolution (over a space or group). Given the particular application, the notion of tensor fields might be more appropriate.
>
> A. Thank you for your suggestion. We replaced the term a "collection of tensors" with a "(discrete) tensor field," which also clarifies the permutation representation.
>
> Q. The authors mention previous work on equivariant neural networks such as steerable convolutions, tensor field networks, and covariant networks, which they claim are inefficient since they use message-passing, while their approach uses GCNs. This contradicts their statement from above, that GCNs are message-passing neural networks. It should be clarified what exactly makes these approaches less efficient.
>
> A. Thank you for the correction. We misused these terms. Our point is that message passing with nonlinear (usually deep) networks is more time-consuming than linear message passing, which is done in GCN (Kipf et al. 2017) and our work. We clarified our point in the paper ( Section 2). In addition, experimental results show that IsoGCN has significant computational efficiency compared to tensor field networks and SE(3)-Transformers. We hope that this is now clarified.
>
> Q. Moreover, since equivariant neural networks are clearly more suitable for the presented experiments then previous GCNs, some of them should be included in the model comparisons with timings.
>
> A. We updated our experiments to include tensor field networks (Thomas et al. 2018) and SE(3)-Transformer (Fuchs et al. 2020), and IsoGCN showed competitive performance against these models with significantly shorter computation time (Section 4 and especially Table 4).
>
> Q. In particular, going to tensors with large rank (as suggested in Eqs. 8-10) should be rather inefficient compared to equivariant networks using spherical harmonics due to the exponentially increasing number of parameters.
>
> A. We added a table summarizing the computation time on tasks outputting the rank-1 tensor and rank-2 tensor (Table 7). As pointed out, the computation time is extended when the rank-2 tensor is output to the rank-1 tensor. However, the magnitude of the increase is not as large as that of the SE(3)-Transformer. Moreover, in Appendix D.4, we added a discussion of the computational complexity of IsoGCN, showing that it increases exponentially regarding the spatial dimension d, which is usually very small (typically, it is 3).
>
> Q. Regarding the timings in Table 4, it is not clear to me whether the timing include the preprocessing of the adjacency matrix for the inference.
>
> A. Thank you for pointing this out. We updated Table 4 to put preprocessing time + inference time because we agree that this is a fairer comparison.
>
> Q. Also the caption mentions that a single CPU core was used. Does this refer to both the neural network and the FEA? Otherwise, it does not seem like fair comparison.
>
> A. Yes, we performed all the computations in Table 4 on a CPU. We have added a clarification regarding this point in the caption of Table 4.
>
> [1] Vojinovic, Marko. "TENSOR CALCULUS." (2010).
> [2] Dullemond, K. "Introduction to Tensor Calculus. Kees Dullemond and Kasper Peeters (1991)."

---

### Author Response · Authors · 2020-11-24
**General comments**

We thank all the reviewers for the feedback and comments. We have uploaded the revised manuscript. The major changes are summarized as follows:
* We updated the introduction section to emphasize the benefit of our results in the application to physical simulations.
* We refined the related work sections to explain the origins of our key ideas, which are GCN (Kipf et al. 2017) and tensor field network (Thomas et al. 2018).
* We cleaned up our notation regarding tensors and IsoAMs. That is a big change and we apologize for any inconvenience. However, we believe that the notation is now more consistent, and the discussion is easier to follow.
* We added tensor field networks (Thomas et al. 2018) and SE(3)-Transformers (Fuchs et al. 2020) as equivariant baseline models in the experiments, showing the clear advantage of IsoGCN in terms of computation time.

We hope that our revised manuscript and responses could address all the questions and uncertainties.

---

### Decision · Program_Chairs · 2021-01-07
**Final Decision**

**Decision:**

Accept (Poster)

**Comment:**

This paper makes an innovative change to the adjacency matrix definition in graph convolutional neural networks (GCNs) (Kipf & Welling, 2017).  The change results in computationally-efficient isometric transformation invariance.   There were a number of concerns raised by reviewers, and the author responses and revisions, and the subsequent discussion, resulted in most of these concerns being satisfactorily addressed.  On reviewer continued to feel the paper was entirely theoretical and therefore not appropriate to ICLR, but that opinion was not shared more broadly and is not held by the area chair.